# Revisiting Frank-Wolfe for Structured Nonconvex Optimization

**Hoomaan Maskan**[*]      **Yikun Hou**[*]      **Suvrit Sra**[†]      **Alp Yurtsever**[*]

[*]Umeå University, Sweden
[†]Technical University of Munich, Germany

## Abstract

We introduce a new projection-free (Frank-Wolfe) method for optimizing structured nonconvex functions that are expressed as a difference of two convex functions. This problem class subsumes smooth nonconvex minimization, positioning our method as a promising alternative to the classical Frank-Wolfe algorithm. DC decompositions are not unique; by carefully selecting a decomposition, we can better exploit the problem structure, improve computational efficiency, and adapt to the underlying problem geometry to find better local solutions. We prove that the proposed method achieves a first-order stationary point in $\mathcal{O}(1/\epsilon^2)$ iterations, matching the complexity of the standard Frank-Wolfe algorithm for smooth nonconvex minimization in general. Specific decompositions can, for instance, yield a gradient-efficient variant that requires only $\mathcal{O}(1/\epsilon)$ calls to the gradient oracle by reusing computed gradients over multiple iterations. Finally, we present numerical experiments demonstrating the effectiveness of the proposed method compared to other projection-free algorithms.

## 1 Introduction

We study projection-free (Frank-Wolfe) algorithms for nonconvex optimization problems of the form:

$$\min_{x \in \mathcal{D}} \quad \phi(x) := f(x) - g(x), \tag{1}$$

where $\mathcal{D} \subseteq \mathbb{R}^d$ is a convex, compact set, and the objective function $\phi : \mathcal{D} \to \mathbb{R}$ is the difference of two convex functions $f : \mathcal{D} \to \mathbb{R}$ and $g : \mathcal{D} \to \mathbb{R}$. We assume that $f$ is $L_f$-smooth on $\mathcal{D}$ (*i.e.*, its gradient is $L_f$-Lipschitz continuous) while $g$ may be nonsmooth.

At first glance, the difference-of-convex (DC) structure may seem like a restriction for nonconvex Frank-Wolfe (FW) methods; but the opposite is true. FW is primarily designed for smooth objectives (Frank & Wolfe, 1956; Jaggi, 2013). Since any smooth function can be expressed as a DC decomposition, our problem template encompasses the minimization of nonconvex smooth functions and extends beyond.

The significance of problem (1) lies in its generality. DC functions form a vector space and constitute a rich subclass of locally Lipschitz nonconvex functions, a class that has received significant attention (Zhang et al., 2020; Kong & Lewis, 2023; Davis et al., 2022). As a result, this problem class has broad applications, including nonconvex quadratic problems (An & Tao, 1997, 1998), convex-concave programming (Yuille & Rangarajan, 2003; Shen et al., 2016), kernel selection (Argyriou et al., 2006), bilevel programming (Hoai An et al., 2009), linear contextual optimization (Bennouna et al., 2024), discrepancy estimation in domain adaptation (Awasthi et al., 2024a), determinantal point processes (Mariet & Sra, 2015), and robustness of neural networks (Awasthi et al., 2024b). We refer to (Le Thi & Pham Dinh, 2018) for a recent survey on the history, development, and applications of DC programming.

39th Conference on Neural Information Processing Systems (NeurIPS 2025).

Our goal in this paper is to develop a flexible projection-free algorithmic framework that preserves the scalability advantages of FW while leveraging the DC structure of the problem. Since DC decompositions are not unique, carefully selecting a decomposition allows us to derive algorithms tailored to address practical limitations in specific problem settings, ultimately leading to improved computational efficiency and better local solutions.

## 1.1 Summary of Contributions

With this background, let us summarize the key contributions of this paper.

1. We design and investigate the convergence behavior of a general projection-free algorithmic framework, which we call *Frank-Wolfe for Difference of Convex problems* (DC-FW). This framework builds on the general template of DC Algorithm (DCA) (Tao & Souad, 1986) and employs FW to solve its subproblems. We show that DC-FW finds an $\epsilon$-suboptimal first-order stationary point in $\mathcal{O}(\epsilon^{-2})$ FW steps, matching the complexity of standard FW for smooth nonconvex minimization (Lacoste-Julien, 2016).

2. DC decomposition for any given function $\phi$ is not unique, and applying DC-FW to different decompositions of the same objective function yields different algorithms. We focus on the general template of $L$-smooth nonconvex minimization over a convex and compact set and examine two natural DC decompositions. The first setting leads to a new nonconvex variant of the conditional gradient sliding algorithm (Lan & Zhou, 2016), which reduces gradient computations by reusing the same gradient over multiple FW steps. As a result, the gradient complexity improves from $\mathcal{O}(\epsilon^{-2})$ to $\mathcal{O}(\epsilon^{-1})$, making it effective for problems where gradient computation is expensive. Moreover, when the problem domain is strongly convex, the linear minimization oracle complexity also improves, from $\mathcal{O}(\epsilon^{-2})$ to $\mathcal{O}(\epsilon^{-3/2})$. The second setting yields an algorithm that follows an inexact proximal point method.

3. Finally, we evaluate the empirical performance of the proposed framework through numerical experiments, comparing it with other FW algorithms on quadratic assignment problems and the alignment of partially observed embeddings.

## 2 Related Work

This section presents related work and background on DC programming and projection-free methods.

## 2.1 DC Algorithm

Many problems in nonconvex optimization can be formulated as a DC program. A widely used approach for solving these problems is the DC Algorithm (DCA), a general framework originally introduced by (Tao & Souad, 1986). DCA has been broadly acknowledged over the past three decades for its wide range of applications. For a comprehensive survey on its variants and applications, we refer to (Le Thi & Pham Dinh, 2018).

The asymptotic convergence theory of DCA was introduced in (Tao, 1997), with a simplified analysis under certain differentiability assumptions later provided in (Lanckriet & Sriperumbudur, 2009). More recently, a non-asymptotic convergence rate of $\mathcal{O}(1/k)$ was established by (Khamaru & Wainwright, 2019; Yurtsever & Sra, 2022; Abbaszadehpeivasti et al., 2023).

## 2.2 Projection-free Methods

In many applications, optimization problems are subject to constraints that impose limits on solutions or act as a form of regularization. When the constraint set $\mathcal{D}$ admits an efficient projection operator, projected gradient methods can be used to solve these problems. However, the projection step can be computationally expensive in many applications. In such settings, projection-free methods, such as the FW algorithm, can offer significant computational advantages.

### 2.2.1 Frank-Wolfe Algorithm

FW was originally proposed by (Frank & Wolfe, 1956) for solving smooth convex minimization problems with polyhedral domain constraints. Its analysis was later extended to arbitrary convex compact sets (Levitin & Polyak, 1966). The algorithm gained recognition for its simple structure and computational efficiency in data science and classical machine learning problems (Jaggi, 2013). The analysis of FW for nonconvex functions was first introduced in (Lacoste-Julien, 2016). Extensions of FW, such as stochastic and block-coordinate versions, as well as variants incorporating away steps,

pairwise steps, and in-face steps for faster convergence, have been extensively studied in the literature. For a comprehensive overview of these developments, we refer to (Kerdreux, 2020).

Khamaru & Wainwright (2019) studied an adaptation of the standard FW algorithm for DC problems, by replacing the gradient in FW with the difference $\nabla f(x) - u$, for some subgradient $u \in \partial g(x)$. They showed that this method finds an $\epsilon$-suboptimal critical point within $\mathcal{O}(1/\epsilon^2)$ iterations. More recently, Millán et al. (2023) studied the same algorithm with a backtracking line-search strategy. For general DC problems, the method with this line-search strategy retains the same complexity guarantee of $\mathcal{O}(1/\epsilon^2)$. However, it achieves a faster rate of $\mathcal{O}(1/\epsilon)$ for a specific class of DC problems where the objective function $\phi$ is weakly-star convex, which means that for every point $x \in \mathcal{D}$, there exists a global optimal point $x_\star$ such that $\phi$ is convex on the line segment between $x$ and $x_\star$.

### 2.2.2 Conditional Gradient Sliding

Lan & Zhou (2016) proposed Conditional Gradient Sliding (CGS) for the constrained minimization of a convex and smooth function. Rather than solving the problem directly with the FW algorithm, CGS formulates an inexact accelerated projected gradient method and employs FW to solve the projection subproblems. While this approach maintains a total of $\mathcal{O}(1/\epsilon)$ FW steps, which matches the complexity of FW for convex minimization, CGS improves the gradient oracle complexity to $\mathcal{O}(1/\sqrt{\epsilon})$, as the gradient remains unchanged within each projection subproblem. Qu et al. (2018) analyzed the convergence of CGS in the nonconvex setting and proved that it requires $\mathcal{O}(1/\epsilon^2)$ FW step and $\mathcal{O}(1/\epsilon)$ gradient evaluations to reach an $\epsilon$-suboptimal stationary point with respect to the squared norm of the gradient mapping.

## 3 Frank-Wolfe for DC Functions

We are now ready to outline the design of our method, which is based on a general class of algorithms for DC problems called the DC Algorithm (DCA). Starting from a feasible initial point $x_0 \in \mathcal{D}$, DCA iteratively updates its estimation by solving the following subproblem:

$$x_{t+1} = \underset{x \in \mathcal{D}}{\operatorname{argmin}} \; \hat{\phi}_t(x) := f(x) - g(x_t) - \langle u_t, x - x_t \rangle, \tag{2}$$

where $u_t \in \partial g(x_t)$ is an arbitrary subgradient of $g$ at $x_t$ Namely, at each iteration, DCA considers a convex surrogate function $\hat{\phi}_t(x)$ obtained by linearizing the concave component around $x_t$.

**Definition 1.** *We measure convergence in terms of the following gap definition:*

$$\operatorname{gap}_{\text{DC}}(x_t) := \max_{x \in \mathcal{D}} \min_{u \in \partial g(x_t)} \left\{ f(x_t) - f(x) - \langle u, x_t - x \rangle \right\}.$$

*When $g$ is differentiable, the subdifferential becomes a singleton, $\partial g(x_t) = \{\nabla g(x_t)\}$, and the gap measure simplifies to $\operatorname{gap}_{\text{DC}}(x_t) = \max_{x \in \mathcal{D}}\{f(x_t) - f(x) - \langle \nabla g(x_t), x_t - x \rangle\}$.*

**Lemma 1.** *The measure $\operatorname{gap}_{\text{DC}}(x_t)$ is nonnegative for any $x_t \in \mathcal{D}$, and it is equal to zero if and only if $x_t$ is a critical point satisfying*

$$(\nabla f(x_t) + \mathcal{N}_{\mathcal{D}}(x_t)) \cap \partial g(x_t) \neq \emptyset, \tag{3}$$

*where $\mathcal{N}_D(x_t)$ is the normal cone of $\mathcal{D}$ at $x_t$.*

*Moreover, if $g$ is differentiable (but not necessarily smooth, i.e., its gradients may not be Lipschitz continuous), then the condition (3) reduces to the characterization of a first-order stationary point. In other words, under the assumption that $g$ is differentiable, $\operatorname{gap}_{\text{DC}}(x_t) = 0$ if and only if $x_t$ is a first-order stationary point of problem (1).*

The following theorem provides convergence guarantees for DCA. Since DCA subproblems may not generally admit a closed-form solution, it is desirable to allow inexact solutions to the DCA subproblem. The theorem ensures that, even when the subproblems are solved inexactly, the method converges to a stationary point (or to a critical point when $g$ is non-differentiable).

**Theorem 2.** *Suppose that the sequence $x_t$ is generated by an inexact-DCA algorithm designed to solve the subproblems described in (2) approximately, satisfying the inequality $\hat{\phi}_t(x_{t+1}) - \min_{x \in \mathcal{D}} \hat{\phi}_t(x) \leq \epsilon/2$ for some $\epsilon > 0$. Then, the following bound holds:*

$$\min_{0 \leq \tau \leq t} \operatorname{gap}_{\text{DC}}(x_\tau) \leq \frac{\phi(x_0) - \phi(x_\star)}{t+1} + \frac{\epsilon}{2}. \tag{4}$$

---

**Algorithm 1** DC-FW

---

1: **Input:** initial point $x_1 \in \mathcal{D}$, target accuracy $\epsilon > 0$
2: **for** $t = 1, 2, \ldots$ **do**
3:     Initialize $X_{t,1} = x_t$
4:     **for** $k = 1, 2, \ldots$ **do**
5:         $S_{t,k} = \arg\min_{x \in \mathcal{D}} \langle \nabla f(X_{t,k}) - \nabla g(x_t), x \rangle$
6:         $D_{t,k} = S_{t,k} - X_{t,k}$
7:         **if** $-\langle \nabla f(X_{t,k}) - \nabla g(x_t), D_{t,k} \rangle \leq \epsilon/2$ **then**
8:            set $x_{t+1} = X_{t,k}$ and break
9:         **end if**
10:        $X_{t,k+1} = X_{t,k} + \eta_{t,k} D_{t,k}$      // use $\eta_{t,k} = 2/(k+1)$, or the strategies in (5) or (6)
11:     **end for**
12: **end for**

---

Note that the subproblem (2) involves smooth convex minimization over a convex compact set, making it suitable for the FW algorithm:

$$s_k = \underset{x \in \mathcal{D}}{\operatorname{argmin}} \ \langle \nabla \hat{\phi}(x_k), x \rangle$$

$$x_{k+1} = x_k + \eta_k(s_k - x_k), \tag{FW}$$

where $\eta_k \in [0, 1]$ is the step-size. There are various step-size strategies in the literature, common choices are $\eta_k = 2/(k+1)$, the exact line-search

$$\eta_k = \underset{\eta \in [0,1]}{\operatorname{argmin}} \ \hat{\phi}\big(x_k + \eta(s_k - x_k)\big), \tag{5}$$

and the Dem'yanov & Rubinov (1970) step-size

$$\eta_k = \min\left\{ \frac{\langle \nabla \hat{\phi}(x_k), x_k - s_k \rangle}{L \|x_k - s_k\|^2}, 1 \right\}. \tag{6}$$

The following result from (Jaggi, 2013) formulates the convergence result for solving these subproblems using the FW algorithm.

**Lemma 3** (Theorem 1 in (Jaggi, 2013))**.** *Suppose $\hat{\phi}$ is a proper, convex, and L-smooth function. Consider the problem of minimizing $\hat{\phi}$ over a convex and compact set $\mathcal{D}$ of diameter $D$. Then, the sequence $\{x_k\}$ generated by the FW algorithm satisfies*

$$\hat{\phi}(x_k) - \hat{\phi}(x) \leq \frac{2LD^2}{k+1}, \qquad \forall x \in \mathcal{D}.$$

*This result holds for $\eta_k = 2/(k+1)$, the line-search, and the Demyanov-Rubinov step-size.*

We are now ready to incorporate DCA and FW algorithms, resulting in a projection-free approach for solving DC problems. The proposed algorithm, DC-FW, applies the inexact-DCA method and employs the FW algorithm to solve the subproblems in (2), as detailed in Algorithm 1.

**Corollary 4.** DC-FW *generates a sequence of solutions that satisfies* $\min_{0 \leq \tau \leq t} \operatorname{gap}_{\mathrm{DC}}(x_\tau) \leq \epsilon$ *within $\mathcal{O}(1/\epsilon)$ iterations, requiring at most $\mathcal{O}(1/\epsilon^2)$ calls to the linear minimization oracle.*

### Improvements for Strongly Convex Sets

Given a DC function $\phi(x) = f(x) - g(x)$, we can always express $\phi$ as a difference of strongly convex functions by adding the same quadratic term to both $f$ and $g$. In this case, subproblem (2) becomes strongly convex, raising a natural question of whether this strong convexity can be exploited. Unfortunately, the FW algorithm generally does not benefit from strong convexity, as worst-case complexity examples also involve strongly convex objective functions (see Lemma 3 in (Jaggi, 2013)). However, if the constraint set $\mathcal{D}$ is also strongly convex, FW achieves a faster convergence rate of $\mathcal{O}(1/k^2)$, as shown by Garber & Hazan (2015).

**Definition 2.** *A set $\mathcal{D} \subseteq \mathbb{R}^d$ is said to be $\alpha$-strongly convex with respect to a norm $\|\cdot\|$ if, for all $x, y \in \mathcal{D}$, all $\eta \in [0,1]$, and all $z \in \mathbb{R}^d$ with $\|z\| \leq 1$, the following inclusion holds:*

$$x + \eta(y - x) + \eta(1 - \eta)\frac{\alpha}{2}\|x - y\|^2 z \in \mathcal{D}.$$

In other words, $\mathcal{D}$ is $\alpha$-strongly convex if, for all $x, y \in \mathcal{D}$ and $\eta \in [0,1]$, it contains a ball of radius $\eta(1 - \eta)\frac{\alpha}{2}\|x - y\|^2$ centered at $x + \eta(y - x)$. Examples of strongly convex sets include the epigraphs of strongly convex functions, as well as $\ell_p$-norm balls and Schatten $p$-norm balls for $1 < p \leq 2$; see (Garber & Hazan, 2015) for details.

**Lemma 5** (Theorem 2 in (Garber & Hazan, 2015))**.** *Suppose $\hat{\phi}$ is a proper, lower semi-continuous, $\mu$-strongly convex and $L$-smooth function. Consider the problem of minimizing $\hat{\phi}$ over an $\alpha$-strongly convex and compact set $\mathcal{D}$ of diameter $D$. Then, the sequence $\{x_k\}$ generated by the FW algorithm satisfies*

$$\hat{\phi}(x_k) - \hat{\phi}(x) \leq \frac{1}{(k+1)^2}\max\left\{\frac{9LD^2}{2}, \frac{48^2 L^2}{\alpha^2 \mu}\right\}, \qquad \forall x \in \mathcal{D}.$$

*This result holds with the line-search or the Demyanov-Rubinov step-size.*

**Corollary 6.** *Consider the DC problem in* (1)*, and assume that $f$ is a strongly convex function and $\mathcal{D}$ is a strongly convex set. Then, the* DC-FW *algorithm generates a sequence of solutions satisfying $\min_{0 \leq \tau \leq t} \mathrm{gap}_{\mathrm{DC}}(x_\tau) \leq \epsilon$ within $\mathcal{O}(1/\epsilon)$ iterations, requiring at most $\mathcal{O}(1/\epsilon^{3/2})$ calls to the linear minimization oracle.*

**Remark 7** (Comparison with (Khamaru & Wainwright, 2019; Millán et al., 2023))**.** *In contrast to our work, these methods do not fully exploit the DC structure of the underlying problem, but rather focus on the classical FW algorithm. While they achieve similar iteration complexity guarantees for the gradient oracle ($\nabla f$) and the linear minimization oracle of order $\mathcal{O}(1/\epsilon^2)$,* DC-FW *benefits from an improved subgradient oracle ($u \in \partial g$) complexity of $\mathcal{O}(1/\epsilon)$. Moreover, if $\mathcal{D}$ is a strongly convex set, our complexity guarantees for both the gradient and linear minimization oracles improve to $\mathcal{O}(1/\epsilon^{3/2})$. Notably, while the linear minimization is typically the primary computational cost in many FW applications within the convex setting, the subgradient computation can also present a significant challenge in DC problems. Additionally, the method proposed by Millán et al. (2023) adopts an iterative line-search strategy that requires function evaluations of $\phi$, incurring an additional computational cost. We evaluate the empirical performance of* DC-FW *against these methods on a partially observed embedding alignment problem in Section 6.2, demonstrating its effectiveness.*

## 4 Special Cases for Smooth Optimization

A particular problem setting for the standard FW algorithm is the minimization of a smooth nonconvex objective function over compact convex set (Lacoste-Julien, 2016):

$$\min_{x \in \mathcal{D}} \phi(x) \tag{7}$$

where $\mathcal{D} \subseteq \mathbb{R}^d$ is a convex and compact set and $\phi : \mathcal{D} \to \mathbb{R}$ is a nonconvex $L$-smooth function. Due to smoothness, both $\phi(x) + \frac{L}{2}\|x\|^2$ and $\frac{L}{2}\|x\|^2 - \phi(x)$ are convex, providing two natural DC decompositions of the objective function, leading to two different algorithms.

### 4.1 Conditional Gradient Sliding

One possible DC decomposition of Problem (7) is:

$$f(x) = \frac{L}{2}\|x\|^2, \quad g(x) = \frac{L}{2}\|x\|^2 - \phi(x). \tag{8}$$

When DCA is applied to this formulation with exact solutions to the subproblems, it recovers the projected gradient method. Specifically, subproblem (2) becomes:

$$\min_{x \in \mathcal{D}} \frac{L}{2}\|x\|^2 - \langle Lx_t - \nabla\phi(x_t), x\rangle. \tag{9}$$

In other words, DC-FW applied to this formulation simplifies to an inexact projected gradient method, where FW is used to approximately solve the subproblems. This naturally leads to a nonconvex variant of the conditional gradient sliding algorithm.

**Remark 8.** *In this setting, the line-search rule (5) is equal to the Demyanov-Rubinov step-size (6).*

**Theorem 9.** *Consider the problem of minimizing an L-smooth (possibly nonconvex) objective function $\phi$ over a convex and compact set $\mathcal{D}$. Suppose we apply DC-FW using the DC decomposition in (8). Then, the sequence $\{x_t\}$ generated by the algorithm satisfies*

$$\min_{0 \leq \tau \leq t} \text{gap}_{\text{DC}}(x_\tau) \leq \frac{\phi(x_0) - \phi(x_\star)}{t+1} + \frac{\epsilon}{2}, \tag{10}$$

*and the inner loop terminates after at most $K \leq 4LD^2/\epsilon$ iterations. As a consequence, the method provably finds an $\epsilon$-suboptimal stationary point after $\mathcal{O}(1/\epsilon)$ gradient evaluations and $\mathcal{O}(1/\epsilon^2)$ linear minimization oracle calls. Moreover, if $\mathcal{D}$ is a strongly convex set, the linear minimization oracle complexity improves to $\mathcal{O}(1/\epsilon^{3/2})$, provided that the Demyanov-Rubinov step-size is used.*

**Lemma 10.** *When applied to the decomposition in (8), $\text{gap}_{\text{DC}}$ provides an upper bound on the widely used projected gradient mapping (PGM) measure:*

$$\text{gap}_{\text{DC}}(x_t) \geq \text{gap}^L_{\text{PGM}}(x_t) := \frac{L}{2} \left\| x_t - \text{proj}_{\mathcal{D}}\left(x_t - \tfrac{1}{L}\nabla\phi(x_t)\right) \right\|^2.$$

**Remark 11** (Comparison with (Qu et al., 2018)). *Our gradient and linear minimization oracle complexities match those established by Qu et al. (2018) for the standard conditional gradient sliding algorithm in the nonconvex setting. However, when adapting the method from the original work of Lan & Zhou (2016), they also inherited the momentum steps. These steps appear redundant in the nonconvex setting in terms of complexity guarantees and complicate the analysis. Our framework simplifies both the method and the analysis while strengthening the guarantees by reducing the constant from $24(\phi(x_0) - \phi(x_\star))$ to $(\phi(x_0) - \phi(x_\star))$, and deriving guarantees in terms of the stronger notion of $\text{gap}_{\text{DC}}$ instead of $\text{gap}_{\text{PGM}}$. Notably, while Qu et al. (2018) derived guarantees based on the gradient mapping norm instead of the standard Frank-Wolfe gap, they explicitly noted that understanding the precise relationship between these convergence criteria was an important direction for future research. We established a clear connection between these convergence criteria.*

### 4.2 Proximal Point Frank-Wolfe

Alternatively, we consider the decomposition:

$$f(x) = \phi(x) + \frac{L}{2}\|x\|^2, \quad g(x) = \frac{L}{2}\|x\|^2. \tag{11}$$

In this setting, subproblem (2) simplifies to

$$\min_{x \in \mathcal{D}} \ \phi(x) + \frac{L}{2}\|x\|^2 - \langle Lx_t, x\rangle.$$

As a result, this decomposition leads to an inexact proximal point algorithm where FW is used to approximately solve the subproblems.

**Theorem 12.** *Consider the problem of minimizing an L-smooth (possibly nonconvex) objective function $\phi$ over a convex and compact set $\mathcal{D}$. Suppose we apply DC-FW using the DC decomposition in (11). Then, the sequence $\{x_t\}$ generated by the algorithm satisfies the bound in (10). In particular, the method finds an $\epsilon$-suboptimal stationary point after $\mathcal{O}(1/\epsilon)$ iterations. This leads to $\mathcal{O}(1/\epsilon^2)$ gradient evaluations and linear minimization oracle calls.*

**Lemma 13.** *When applied to the decomposition in (11), $\text{gap}_{\text{DC}}$ provides an upper bound on the widely used proximal point mapping (PPM) measure:*

$$\text{gap}_{\text{DC}}(x_t) \geq \text{gap}^L_{\text{PPM}}(x_t) := \frac{L}{2}\left\| x_t - \text{prox}_{\frac{1}{L}\phi}(x_t)\right\|^2.$$

**On the role of the decomposition.** It is important to note that $\text{gap}_{\text{DC}}$ is decomposition-dependent. The choice of DC decomposition determines the notion of stationarity captured by the gap function. Consequently, it allows one to trade off computational efficiency against the strength of the stationarity notion achieved. Recall from Definition 1 that $\text{gap}_{\text{DC}}$ retains the convex component $f$ while linearizing the concave part $-g$, effectively discarding higher-order information in $g$. To illustrate this effect, consider the smooth nonconvex function $\phi(x) = \sin(\pi x_1)\cos(\pi x_2)$ over the domain $[-1, 1]^2$ with

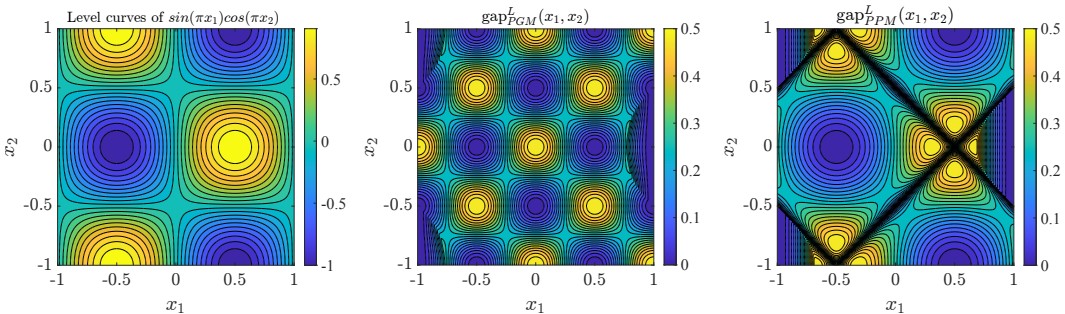

Figure 1: Comparison of the DC gap function for different decompositions of $\phi(x_1, x_2) = \sin(\pi x_1) \cos(\pi x_2)$ on the domain $[-1, 1]^2$. *[Left]* Level curves of $\phi$. *[Middle]* $\mathrm{gap}_{\mathrm{PGM}}^L$, corresponding to the decomposition in Section 4.1, which linearizes $\phi$ and therefore does not distinguish between local minima, saddle points, and local maxima. *[Right]* $\mathrm{gap}_{\mathrm{PPM}}^L$, corresponding to the decomposition in Section 4.2, which retains curvature information in $\phi$; it is flatter around local minima and sharper around saddle points and local maxima.

$L = \pi^2$. We evaluate both $\mathrm{gap}_{\mathrm{PGM}}^L$ and $\mathrm{gap}_{\mathrm{PPM}}^L$ over a fine grid on the domain. The results, shown in Figure 1, reveal that while $\mathrm{gap}_{\mathrm{PGM}}^L$ behaves similarly across all stationary points (whether they are local minima, saddle points, and local maxima), $\mathrm{gap}_{\mathrm{PPM}}^L$ distinguishes between them: it is flatter near the local minimum but sharper near the saddle and local maximum. This difference arises because $\mathrm{gap}_{\mathrm{PPM}}^L$ retains the curvature of $\phi$ within $f$, whereas $\mathrm{gap}_{\mathrm{PGM}}^L$ linearizes it away.

## 5 Special Cases for Nonsmooth Optimization

We now extend the discussion to a class of nonsmooth problems that fit into the proposed framework.

### 5.1 Conditional Subgradient Sliding

Consider an objective function $\phi : \mathcal{D} \to \mathbb{R}$, which may be nonsmooth and nonconvex, but whose negative is $\omega$-weakly convex; that is, $-\phi(x) + \frac{\omega}{2}\|x\|^2$ is convex. Then the following DC decomposition is valid:

$$f(x) = \frac{\omega}{2}\|x\|^2, \qquad g(x) = \frac{\omega}{2}\|x\|^2 - \phi(x).$$

Unlike in Section 4.1, $\phi$ is not required to be smooth (or even continuously differentiable), it suffices that a subgradient of $g$ (equivalently, of $-\phi$) can be computed. Under this formulation, DC-FW naturally extends the conditional gradient sliding method to the nonsmooth setting. The result in Theorem 9 continues to hold in this setting after replacing the smoothness constant $L$ with the weak convexity constant $\omega$, and gradient evaluations with subgradient evaluations.

A notable subclass arises when $\phi$ admits the composite form

$$\phi(x) = p(x) - q(x), \tag{12}$$

where $p$ is $\omega$-smooth (possibly nonconvex) and $q$ is convex (possibly nonsmooth). In this case, $-\phi$ is $\omega$-weakly convex and $g$ admits a well-defined subgradient. Applying DC-FW to this formulation economizes on both gradient evaluations of $p$ and subgradient computations of $q$. As an example, the alignment problem of partially observed embeddings presented in Section 6.2 belongs to this class, characterized by a convex smooth loss term and concave nonsmooth regularizers.

### 5.2 Proximal Subgradient Frank-Wolfe

Consider the composite objective in (12), but using the following DC decomposition:

$$f(x) = p(x) + \frac{\omega}{2}\|x\|^2, \qquad g(x) = \frac{\omega}{2}\|x\|^2 + q(x).$$

This is a valid decomposition for DC-FW, since both $f$ and $g$ are convex, and $f$ is $2\omega$-smooth. The resulting algorithm can be interpreted as an inexact proximal subgradient method with FW for solving each subproblem approximately. Similar to the result in Theorem 12, the method attains an $\epsilon$-suboptimal stationary point after $\mathcal{O}(1/\epsilon)$ iterations. However, this requires $\mathcal{O}(1/\epsilon)$ subgradient evaluations of $q$, and $\mathcal{O}(1/\epsilon^2)$ gradient evaluations of $p$ and linear minimization oracle calls.

# 6 Numerical Experiments

In this section, we numerically evaluate DC-FW on solving the quadratic assignment problem (QAP) and alignment of partially observed embeddings. Simulations were run on a single core of an Intel Xeon Gold 6132 processor using MATLAB 2021a.

## 6.1 Quadratic Assignment Problem

The goal in QAP is to find a permutation that optimally aligns two matrices $A$ and $B \in \mathbb{R}^{n \times n}$. This amounts to minimizing a nonconvex quadratic objective function over the set of permutation matrices, an NP-Hard combinatorial optimization problem. A promising approach to approximate QAP is to use the following convex-hull relaxation (Vogelstein et al., 2015):

$$\min_{X \in \mathbb{R}^{n \times n}} \langle A^\top X, XB \rangle \quad \text{subj.to} \quad X \in [0, 1]^{n \times n}, \ X1_n = X^\top 1_n = 1_n, \tag{13}$$

where $1_n$ denotes the $n$-dimensional vector of ones. The feasible set in this problem, known as the Birkhoff polytope, is the convex hull of permutation matrices. In general, this relaxation is still nonconvex due the quadratic objective function. The relax-and-round strategy of Vogelstein et al. (2015) involves two main steps: Finding a local optimal solution of (13) and rounding it to the closest permutation matrix.

Projecting a matrix onto the Birkhoff polytope is computationally expensive. Therefore, Vogelstein et al. (2015) employs the FW algorithm for solving (13). The linear minimization oracle for this problem corresponds to solving linear assignment problem (LAP), which can be solved $\mathcal{O}(n^3)$ arithmetic operations by using the Hungarian method or the Jonker-Volgenant algorithm (Kuhn, 1955; Munkres, 1957; Jonker & Volgenant, 1987). After finding a solution to the relaxed problem (13), we can round it to the closest permutation matrix (in Frobenius norm) also by solving a LAP.

**DC-FW for solving QAP.** We consider the decomposition $\phi(X) = f(X) - g(X)$ with the components

$$f(X) = \frac{1}{4}\|A^\top X + XB\|_F^2, \quad g(X) = \frac{1}{4}\|A^\top X - XB\|_F^2. \tag{14}$$

While other decompositions, such as those in (8) and (11) are also possible, here we show the results for (14) which performed the best. The numerical results for other decompositions and the non-convex CGS method (Qu et al., 2018) are given in Appendix B.

**Evaluation metric.** To compare the solutions of DC-FW and FW (as used in (Vogelstein et al., 2015)), we evaluate the assignment error from rounded permutation matrices:

$$\text{assignment error} = \frac{\phi(\hat{X}_\tau) - \phi(\hat{X}_{\text{best}})}{\max\{\phi(\hat{X}_{\text{best}}), 1\}}, \tag{AE}$$

where $\hat{X}_{\text{best}}$ denotes the best known solution to QAP, which is available for the QAPLIB benchmark datasets (Burkard et al., 1997).

**Implementation details.** All methods start from the same initial point, obtained by projecting the sum of a normalized matrix of ones and an *i.i.d.* standard Gaussian random matrix onto the Birkhoff polytope. This projection is performed using $10^3$ iterations of the alternating projections method. We compute the gap at the initial point, denoted as $\epsilon_0$, and terminate the algorithms once the gap reaches $0.001 \times \epsilon_0$. Additionally, we impose a maximum iteration limit of $10^8$. For both DC-FW and FW, we used the exact line-search method for step-size selection, see Appendix B for more details. The inner loop for DC-FW terminates with respect to a specific tolerance level $\epsilon$. We employed an adaptive strategy where the tolerance $\epsilon_t$ is updated dynamically. We fixed a multiplication parameter $\beta = 0.8$, and initially we set $\epsilon_1 = \beta \times \epsilon_0$. Then, at each iteration $t$, we update the tolerance as follows: if the gap falls below $\epsilon_t$, we decrease the tolerance by setting $\epsilon_{t+1} = \beta \epsilon_t$. Otherwise, we keep it unchanged, $\epsilon_{t+1} = \epsilon_t$. Further details and discussion on this strategy are provided in Appendix D.

**Results.** Figure 2 demonstrate the superior performance of DC-FW over the FW method on 73 out of 134 datasets in terms of the assignment error. In 18 cases, both methods produced equal errors, while FW achieved a lower error on 43 datasets. Moreover, DC-FW achieved a lower average assignment error (0.0085) compared to FW (0.0112).

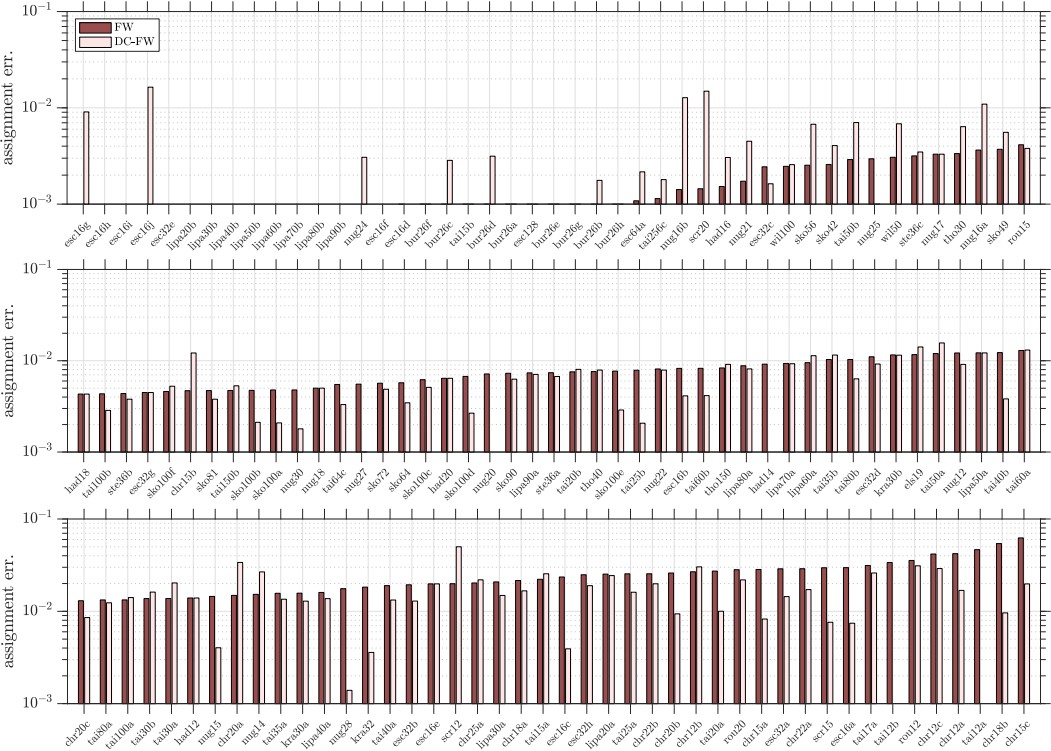

Figure 2: Assignment error of FW and Dc-Fw for solving QAP using relax-and-round strategy. Zero shows an exact solution. The instances are ordered from best to worst performance of FW. In total, 134 datasets from QAPLIB were used: Dc-Fw outperformed FW in 73 cases, FW performed better in 43 cases, and both methods achieved the same assignment error in 18 cases.

## 6.2 Alignment of Partially Observed Embeddings

We consider a cross-lingual word embedding alignment problem, where the goal is to align the source embedding matrix ($E_1 \in \mathbb{R}^{d \times n}$) with the target embedding matrix ($E_2 \in \mathbb{R}^{d \times n}$) using an orthogonal transformation matrix $X \in \mathbb{R}^{d \times d}$. When the embedding matrices are fully available, this problem can be efficiently solved using the Procrustes algorithm (Xing et al., 2015; Conneau et al., 2017). Here, we consider a more challenging setting where the target embedding is partially observed. In this scenario, we are given a measurement mask $P \in \{0, 1\}^{d \times n}$ and the partial observations $Y = P \odot E_2 \in \mathbb{R}^{d \times n}$, where $\odot$ denotes the Hadamard product. Our goal is to find an orthogonal transformation that minimizes the mean squared loss. We consider the following objective with a nonconvex regularizer that promotes orthogonality:

$$\min_{X \in \mathbb{R}^{d \times d}} \frac{1}{2n}\|P \odot (XE_1) - Y\|_F^2 - \lambda\|X\|_* \quad \text{subj.to} \quad \|X\| \leq 1. \qquad (15)$$

Here, the spectral norm constraint ensures that the singular values of $X$ are bounded by 1, and the negative nuclear norm regularization encourages them to approach this limit, promoting orthogonality.

We compare the performance of Dc-Fw with other FW variants for DC problems as described in (Khamaru & Wainwright, 2019) and (Millán et al., 2023), which we denote by FW-K and FW-M respectively. A subgradient of nuclear norm can be found as $UV^\top$, where $U$ and $V$ are the left and right singular vector matrices. Similarly, a linear minimization oracle over the spectral norm can be obtained as $-UV^\top$. Additionally, FW-M requires evaluating the objective function during its line-search, which involves calculating the nuclear norm, which has a similar cost (i.e., SVD computation). While we implemented the oracles using full SVD computations in this experiment, the approximation $UV^\top$ can be obtained using a few iterations of the Newton-Schulz method (Bernstein & Newhouse, 2024) in large-scale implementations. However, it is worth noting that implementing FW-M with inexact oracles can be challenging, as the backtracking line-search procedure may enter an infinite loop when the objective evaluations are inaccurate.

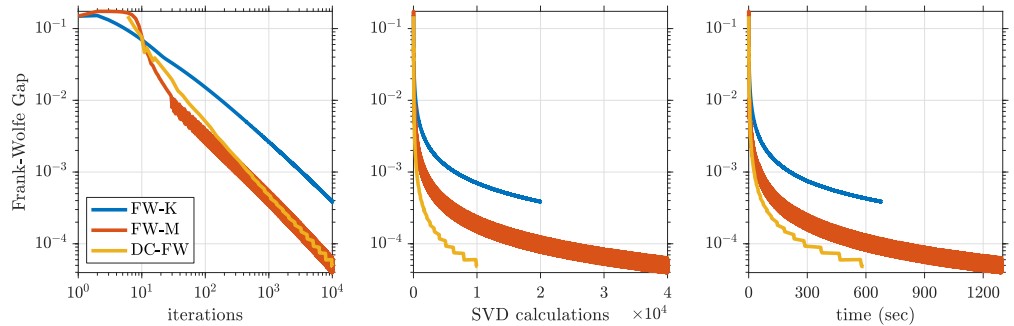

Figure 3: FW gap evolution as a function of the iteration counter (left), the number of SVD computations (middle), and the wall clock time (right) for the alignment of partially observed embeddings. In $10^4$ iterations, FW-K and FW-M called the subgradient and linear minimization oracles $10^4$ times each; FW-M performed an additional $19,990$ function evaluations during the backtracking line-search; DC-FW called the linear minimization $10^4$ times and the subgradient $88$ times.

We use the 300-dimensional English and French word embeddings from the FastText library (Bojanowski et al., 2017) as the source and target embeddings, selecting the 10,000 most common words from each dictionary. We model partial observations by assuming that each entry in the target embedding is independently observed with a probability of $0.1$. We choose regularization parameter $\lambda = 10^{-4}$. All algorithms are initialized at the origin. Figure 3 presents the results of this experiment. Both FW-M and DC-FW outperform FW-K. While the iteration convergence rates of FW-M and DC-FW appear similar, the iterations of FW-M are more expensive because they require computing the subgradient at every iteration and evaluating the objective function within the backtracking line-search procedure. FW-M also appears to oscillate more.

To evaluate the quality of the obtained solution, we use the baseline alignment quality computed by assuming that $E_2$ is fully available and solving the problem via the Procrustes algorithm. After rounding our solutions to the closest orthogonal matrix, we compare the achieved alignment quality with the baseline. Our formulation and method achieves $93.11\%$ of the original alignment quality despite observing only $10\%$ of the target embedding, demonstrating that the formulation is effective even under partial observations.

## 7 Conclusions

We studied the DC-FW framework, which integrates DCA with the FW algorithm. We established that DC-FW retains the same oracle complexity for linear minimization steps as standard FW in nonconvex optimization. However, by carefully selecting the DC decomposition, DC-FW can adapt to the underlying problem geometry in different ways, offering a flexible approach for a variety of problem settings. We examined two natural DC decompositions within the standard smooth minimization template. One decomposition, in particular, led to an algorithm that reduces gradient computations, improving the gradient complexity from $\mathcal{O}(\epsilon^{-2})$ to $\mathcal{O}(\epsilon^{-1})$. Additionally, when the problem domain is strongly convex, this approach improves the linear minimization oracle complexity to $\mathcal{O}(\epsilon^{-3/2})$. Through numerical experiments on quadratic assignment problem and alignment of partially observed embeddings, we demonstrated the effectiveness of DC-FW in comparison to the standard FW approach. After the initial submission of this paper, Pokutta (2025) presented an extensive computational study that builds upon our framework, combining our approach with some advanced FW variants (for example, with blended pairwise steps (Tsuji et al., 2022)). Their results provide complementary evidence for the scalability and efficiency of the proposed approach.

Our findings provide a concrete example where a simple but deliberate manipulation of the DC decomposition enables a better algorithm, and calls for a wider future study. The simplicity of our analysis in achieving these results shows the potential of problem reformulations within the DC framework. Rather than relying on intricate analytical techniques, we overcome technical challenges through carefully chosen problem reformulations. Our work also suggests noteworthy future directions. Extending the analysis of DC-FW to stochastic settings and making it more compatible with modern machine learning practices, remains a crucial next step. Another potential avenue is to develop adaptive DC decompositions that dynamically adjust to the problem's geometry.

## Acknowledgments and Disclosure of Funding

Hoomaan Maskan, Yikun Hou, and Alp Yurtsever were supported by the Wallenberg AI, Autonomous Systems, and Software Program (WASP), funded by the Knut and Alice Wallenberg Foundation. Suvrit Sra acknowledges generous support from the Alexander von Humboldt Foundation. Part of the computations was conducted using resources from the High Performance Computing Center North (HPC2N) and were enabled by the National Academic Infrastructure for Supercomputing in Sweden (NAISS), partially funded by the Swedish Research Council under grant no. 2022-06725. The remaining computations were performed on the Berzelius resource, provided by the Knut and Alice Wallenberg Foundation at the National Supercomputer Centre. We thank Ahmet Alacaoglu and Yura Malitsky for insightful discussions on the relationships between different gap functions.

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

# A Proofs and Technical Details

## A.1 Proof of Lemma 1

*Proof.* The first statement is straightforward, since $x_t \in \mathcal{D}$, and choosing $x = x_t$ within the maximization gives zero.

Suppose we have $\mathrm{gap}_{\mathrm{DC}}(x_t) = 0$. Then, there exists a subgradient $u_t^\star \in \partial g(x_t)$ such that

$$f(x) - f(x_t) - \langle u_t^\star, x - x_t \rangle \geq 0, \quad \forall x \in \mathcal{D}.$$

Consider $x = x_t + \alpha d$ for an arbitrary feasible direction $d$ and step-size $\alpha > 0$. Then,

$$f(x_t + \alpha d) - f(x_t) - \langle u_t^\star, \alpha d \rangle \geq 0 \quad \forall \alpha d : x_t + \alpha d \in \mathcal{D}.$$

Dividing by $\alpha$ and taking limit as $\alpha \to 0^+$, we obtain

$$\langle \nabla f(x_t), d \rangle - \langle u_t^\star, d \rangle \geq 0 \quad \forall d : \lim_{\alpha \to 0^+} x_t + \alpha d \in \mathcal{D}.$$

Since $\mathcal{D}$ is closed and convex, $\lim_{\alpha \to 0^+} x_t + \alpha d \in \mathcal{D}$ for any $d = x - x_t$ such that $x \in \mathcal{D}$. Consequently,

$$\langle \nabla f(x_t) - u_t^\star, x_t - x \rangle \leq 0 \quad \forall x \in \mathcal{D},$$

or, equivalently, $u_t^\star - \nabla f(x_t) \in \mathcal{N}_{\mathcal{D}}(x_t)$. Therefore, we have condition (3) satisfied.

Next, suppose condition (3) is satisfied. Then, there exists a subgradient $u_t^* \in \partial g(x_t)$ such that $u_t^* - \nabla f(x_t) \in \mathcal{N}_{\mathcal{D}}(x_t)$, which means

$$\langle \nabla f(x_t) - u_t^*, x_t - x \rangle \leq 0 \quad \forall x \in \mathcal{D}.$$

Since $f$ is convex, it follows that

$$f(x_t) - f(x) - \langle u_t^*, x_t - x \rangle \leq 0 \quad \forall x \in \mathcal{D}.$$

Since this inequality holds for all $x \in \mathcal{D}$, we can maximize the left-hand side over $x \in \mathcal{D}$. Moreover, since it holds for at least one subgradient $u_t^* \in \partial g(x_t)$, we can minimize over the subdifferential set:

$$\min_{u \in \partial g(x_t)} \max_{x \in \mathcal{D}} \left\{ f(x_t) - f(x) - \langle u, x_t - x \rangle \right\} \leq 0.$$

By Von Neumann's minimax theorem, we can exchange the order of $\min$ and $\max$ since (i) $\partial g(x_t)$ and $\mathcal{D}$ are convex and compact sets; and (ii) the expression inside the braces is concave with respect to $x$ for any fixed $u \in \partial g(x_t)$, and convex (in fact, linear) with respect to $u$ for any fixed $x \in \mathcal{D}$. After this change, we obtain $\mathrm{gap}_{\mathrm{DC}}(x_t) \leq 0$. Since the gap measure is nonnegative by definition, we conclude that $\mathrm{gap}_{\mathrm{DC}}(x_t) = 0$. $\qquad\square$

## A.2 Proof of Theorem 2

*Proof.* By convexity of $g$, we have

$$f(x_{t+1}) - g(x_{t+1}) \leq f(x_{t+1}) - g(x_t) - \langle u_t, x_{t+1} - x_t \rangle,$$

where $u_t \in \partial g(x_t)$ is the subgradient used at iteration $t$ of the algorithm. Then, by the update rule (2), the following bound holds $\forall x \in \mathcal{D}$:

$$f(x_{t+1}) - g(x_{t+1}) \leq f(x) - g(x_t) - \langle u_t, x - x_t \rangle + \frac{\epsilon}{2}.$$

Then, we add $f(x_t)$ to both sides and rearrange as follows:

$$f(x_t) - f(x) - \langle u_t, x_t - x \rangle \leq \Big( f(x_t) - g(x_t) \Big) - \Big( f(x_{t+1}) - g(x_{t+1}) \Big) + \frac{\epsilon}{2}.$$

Since this inequality holds for all $x \in \mathcal{D}$, we can maximize the left-hand side over $x \in \mathcal{D}$. Moreover, since it holds for at least one subgradient $u_t^* \in \partial g(x_t)$, we can minimize over the subdifferential set:

$$\min_{u \in \partial g(x_t)} \max_{x \in \mathcal{D}} \left\{ f(x_t) - f(x) - \langle u_t, x_t - x \rangle \right\} \leq \Big( f(x_t) - g(x_t) \Big) - \Big( f(x_{t+1}) - g(x_{t+1}) \Big) + \frac{\epsilon}{2}.$$

By Von Neumann's minimax theorem, we can exchange the order of $\min$ and $\max$, and get

$$\mathrm{gap}_{\mathrm{DC}}(x_t) \leq \Big( f(x_t) - g(x_t) \Big) - \Big( f(x_{t+1}) - g(x_{t+1}) \Big) + \frac{\epsilon}{2}.$$

Finally, we obtain (4) by taking the average of this inequality over $t$ and noting that the minimum is less than or equal to the average. $\qquad\square$

### A.3 Proof of Corollary 4

*Proof.* The first results is an immediate consequence of Theorem 2. Moreover, by Lemma 3, the condition for FW is achieved at most after $K \leq 4LD^2/\epsilon$ iterations. Therefore, the total number of calls to the linear minimization oracle is bounded by $tK \leq \mathcal{O}(1/\epsilon^2)$. $\square$

### A.4 Proof of Corollary 6

*Proof.* The first results is an immediate consequence of Theorem 2. Moreover, by Lemma 5, the condition for FW is achieved at most after $K \leq \max\{3\sqrt{L}D, \frac{48\sqrt{2}L}{\alpha\sqrt{\mu}}\}/\sqrt{\epsilon}$ iterations. Therefore, the total number of calls to the linear minimization oracle is bounded by $tK \leq \mathcal{O}(1/\epsilon^{3/2})$. $\square$

### A.5 Proof of Theorem 9

*Proof.* The first results immediately follows from Theorem 2. By Lemma 3, the $\epsilon/2$-accuracy for FW is achieved at most after $K \leq 4LD^2/\epsilon$ iterations. Therefore, the total number of calls to the linear minimization oracle is bounded by $tK \leq \mathcal{O}(1/\epsilon^2)$.

When $\mathcal{D}$ is an strongly convex set, since (9) is a strongly convex problem with a strongly convex compact constraint set, by Lemma 5, the $\epsilon/2$-accuracy for FW is achieved at most after $K \leq \max\{3\sqrt{L}D, \frac{48\sqrt{2}L}{\alpha\sqrt{\mu}}\}/\sqrt{\epsilon}$ iterations provided that Demyanov-Rubinov step-size is used. Therefore, the total number of calls to the linear minimization oracle is bounded by $tK \leq \mathcal{O}(1/\epsilon^{3/2})$. $\square$

### A.6 Proof of Lemma 10

*Proof.* If we substitute the terms in (8) into the definition of $\mathrm{gap}_{\mathrm{DC}}$, we get

$$\max_{x \in \mathcal{D}} \left\{ \frac{L}{2}\|x_t\|^2 - \frac{L}{2}\|x\|^2 - \langle Lx_t - \nabla\phi(x_t), x_t - x \rangle \right\}$$

$$= \max_{x \in \mathcal{D}} \left\{ \langle \nabla\phi(x_t), x_t - x \rangle - \frac{L}{2}\|x - x_t\|^2 \right\}$$

$$= \langle \nabla\phi(x_t), x_t - x_t^\star \rangle - \frac{L}{2}\|x_t^\star - x_t\|^2$$

where $x_t^\star = \mathrm{proj}_{\mathcal{D}}(x_t - \frac{1}{L}\nabla\phi(x_t))$. By the variational inequality characterization of projection, we have

$$\langle x_t - \frac{1}{L}\nabla\phi(x_t) - x_t^\star, x - x_t^\star \rangle \leq 0, \quad \forall x \in \mathcal{D}.$$

In particular, using this with $x = x_t$, we get

$$\frac{L}{2}\|x_t - x_t^\star\|^2 \leq \langle \nabla\phi(x_t), x_t - x_t^\star \rangle - \frac{L}{2}\|x_t^\star - x_t\|^2 = \mathrm{gap}_{\mathrm{DC}}(x_t). \quad \square$$

### A.7 Proof of Theorem 12

*Proof.* The first results immediately follows from Theorem 2. Since the subproblems are convex, by Lemma 3, the $\epsilon/2$ accuracy for FW is achieved at most after $K \leq 4LD^2/\epsilon$ iterations. Therefore, the total number of calls to the linear minimization oracle is bounded by $tK \leq \mathcal{O}(1/\epsilon^2)$. $\square$

### A.8 Proof of Lemma 13

*Proof.* Plugging the terms from (11) into the definition of $\mathrm{gap}_{\mathrm{DC}}$ gives

$$\max_{x \in \mathcal{D}} \left\{ \phi(x_t) + \frac{L}{2}\|x_t\|^2 - \phi(x) - \frac{L}{2}\|x\|^2 - \langle Lx_t, x_t - x \rangle \right\}$$

$$= \max_{x \in \mathcal{D}} \left\{ \phi(x_t) - \phi(x) - \frac{L}{2}\|x - x_t\|^2 \right\}$$

$$= \phi(x_t) - \phi(x_t^\star) - \frac{L}{2}\|x_t - x_t^\star\|^2$$

where $x_t^\star = \operatorname{prox}_{\frac{1}{L}\phi}(x_t)$. The variational inequality characterization of the proximal operator gives

$$\langle x_t - x_t^\star, x - x_t^\star \rangle \leq \frac{1}{L}\phi(x) - \frac{1}{L}\phi(x_t^\star), \quad \forall x \in \mathcal{D}.$$

In particular, applying this at $x = x_t$, we obtain

$$\frac{L}{2}\|x_t - x_t^\star\|^2 \leq \phi(x_t) - \phi(x_t^\star) - \frac{L}{2}\|x^t - x_t^\star\|^2 = \operatorname{gap}_{\mathrm{DC}}(x_t). \qquad \square$$

# B  Additional Details on the QAP Experiments

## B.1  Exact Line-Search

Here, we derive the exact line-search step. We consider the decomposition $\phi(X) = f(X) - g(X)$ with three different variants of components

$$f(X) = \frac{1}{4}\|A^\top X + XB\|_F^2 \qquad g(X) = \frac{1}{4}\|A^\top X - XB\|_F^2. \qquad \text{(variant 1)}$$

$$f(x) = \frac{L}{2}\|X\|_F^2 \qquad g(x) = \frac{L}{2}\|X\|_F^2 - \langle A^\top X, XB\rangle. \qquad \text{(variant 2)}$$

$$f(x) = \langle A^\top X, XB\rangle + \frac{L}{2}\|X\|_F^2 \qquad g(x) = \frac{L}{2}\|X\|_F^2 \qquad \text{(variant 3)}$$

### Variant 1

We can compute the gradients for $f$ and $g$ by

$$\nabla f(X) = \frac{1}{2}\left(A(A^\top X + XB) + (A^\top X + XB)B^\top\right)$$

$$\nabla g(X) = \frac{1}{2}\left(A(A^\top X - XB) - (A^\top X - XB)B^\top\right).$$

For the line-search in (5), we get

$$\frac{d}{d\eta}\hat{\phi}_t(X_{tk} + \eta D_{tk}) = \frac{d}{d\eta}f(X_{tk} + \eta D_{tk}) - \frac{d}{d\eta}\langle \nabla g(X_t), X_{tk} + \eta D_{tk} - X_{tk}\rangle$$

$$= \frac{1}{4}\frac{d}{d\eta}\|(A^\top X_{tk} + X_{tk}B) + \eta(A^\top D_{tk} + D_{tk}B)\|_F^2 - \langle \nabla g(X_t), D_{tk}\rangle$$

$$= \frac{1}{2}\langle A^\top D_{tk} + D_{tk}B, (A^\top X_{tk} + X_{tk}B) + \eta(A^\top D_{tk} + D_{tk}B)\rangle - \langle \nabla g(X_t), D_{tk}\rangle$$

$$= \frac{1}{2}\eta\|A^\top D_{tk} + D_{tk}B\|_F^2 + \frac{1}{2}\langle A^\top D_{tk} + D_{tk}B, A^\top X_{tk} + X_{tk}B\rangle - \langle \nabla g(X_t), D_{tk}\rangle$$

Equating this to 0, we get

$$\eta = \frac{2\langle \nabla g(X_t), D_{tk}\rangle - \langle A^\top D_{tk} + D_{tk}B, A^\top X_{tk} + X_{tk}B\rangle}{\|A^\top D_{tk} + D_{tk}B\|_F^2}.$$

### Variant 2

We can solve (13) using DC-FW for

$$f(x) = \frac{L}{2}\|X\|_F^2, \qquad g(x) = \frac{L}{2}\|X\|_F^2 - \langle A^\top X, XB\rangle, \qquad (16)$$

Now, the gradients for $f$ and $g$ become

$$\nabla f(X) = LX$$

$$\nabla g(X) = LX - A^\top XB^\top - AXB.$$

Utilizing the gradients in line-search (5) we get

$$\frac{d}{d\eta}\hat{\phi}_t(X_{tk} + \eta D_{tk}) = \frac{d}{d\eta}f(X_{tk} + \eta D_{tk}) - \frac{d}{d\eta}\langle \nabla g(X_t), X_{tk} + \eta D_{tk} - X_{tk}\rangle$$

$$= \langle D_{tk}, L(X_{tk} + \eta D_{tk}))\rangle - \langle \nabla g(X_t), D_{tk}\rangle$$

Equating this to 0 gives

$$\eta = \frac{\langle \nabla g(X_t), D_{tk}\rangle - L\langle D_{tk}, X_{tk}\rangle}{L\|D_{tk}\|_F^2}.$$

**Variant 3**

We can solve (13) using DC-FW for

$$f(x) = \langle A^\top X, XB \rangle + \frac{L}{2}\|X\|_F^2, \qquad g(x) = \frac{L}{2}\|X\|_F^2. \tag{17}$$

Now, the gradients for $f$ and $g$ become

$$\nabla f(X) = A^\top X B^\top + AXB + LX$$
$$\nabla g(X) = LX.$$

Utilizing the gradients in line-search (5) we get

$$\frac{d}{d\eta}\hat{\phi}_t(X_{tk} + \eta D_{tk}) = \frac{d}{d\eta}f(X_{tk} + \eta D_{tk}) - \frac{d}{d\eta}\langle \nabla g(X_t), X_{tk} + \eta D_{tk} - X_{tk}\rangle$$

$$= \langle D_{tk}, (A^\top(X_{tk} + \eta D_{tk}))B^\top + A(X_{tk} + \eta D_{tk})B + L(X_{tk} + \eta D_{tk}))\rangle - \langle \nabla g(X_t), D_{tk}\rangle$$

$$= \langle D_{tk}, A^\top X_{tk}B^\top + AX_{tk}B + LX_{tk} + \eta(A^\top D_{tk}B^\top + AD_{tk}B + LD_{tk})\rangle - \langle \nabla g(X_t), D_{tk}\rangle$$

Equating this to 0 gives

$$\eta = \frac{\langle \nabla g(X_t), D_{tk}\rangle - \langle D_{tk}, A^\top X_{tk}B^\top + AX_{tk}B + LX_{tk}\rangle}{2\langle A^\top D_{tk}, D_{tk}B\rangle + L\|D_{tk}\|_F^2}$$

### B.1.1 Numerical Results

We have conducted additional experiments to emphasize the superiority of DC-FW in finding better stationary solutions. We denote DC-FW (var 2) as in (16) and DC-FW (var 2) as in (17).

Further, we implemented the Non-convex CGS as explained in the (Qu et al., 2018). According to the original work, if the CGS is called $T$ times, then the inner loop requires an accuracy of $\mathcal{O}(1/T)$ for the subproblems. This, however, takes a very long time. Therefore, we used a similar technique to the DC-FW implementation to both improve the accuracy and the speed of Non-convex CGS.

For all the methods similar stopping criteria (as explained in Section 6) was used and line search method determined the step-sizes. Additionally, a stop-time of 10 hours was considered for all the implementations. We repeated 10 Monte-Carlo instances per method per dataset. The results are summarized in Figure 4.

The heatmap shows the average number of datasets with lower assignment error for each method in the y-axis against each method on the x-axis. 90% confidence intervals were used to show the statistical significance of the experiments. For example, DCFW (var 1) performs better than the CGS method in 68.70 cases on average with 90% confidence interval [66.17, 71.23]. On the other hand, CGS did better than DCFW (var 1) in 46.40 cases on average with 90% confidence interval [43.76, 49.04]. The table shows the superiority of DC-FW type methods (all variations) against CGS and the FW methods.

### B.2 Projection vs Linear Minimization for the Birkhoff Polytope

Consider for instance the set of $(n \times n)$ doubly stochastic matrices (*i.e.,* the convex hull of permutation matrices, also known as the Birkhoff polytope). This constraint frequently appears in allocation problems like optimal transport or quadratic assignment problems. To our knowledge, no polynomial-time exact solution method is known for this projection. An $\epsilon$-precise approximation can be computed via iterative methods, such as operator splitting methods like Douglas-Rachford splitting, incurring an arithmetic cost of $\mathcal{O}(n^3/\epsilon^2)$ (Combettes & Pokutta, 2021; Bertsekas, 1992). In contrast, an exact solution to the linear minimization oracle of FW can be found by using the Hungarian method or the Jonker-Volgenant algorithm at $\mathcal{O}(n^3)$ arithmetic operations (Kuhn, 1955; Munkres, 1957; Jonker & Volgenant, 1987). Alternatively, an $\epsilon$-precise approximation can be achieved in $\mathcal{O}(n^2/\epsilon)$ operations with the auction algorithm (Alfaro et al., 2022; Bertsekas, 1992).

## C Additional Numerical Experiments on Neural Network Training

**Classification on Neural Networks**   We used a heuristic stochastic variant of the proposed DC-FW algorithm and the stochastic Frank-Wolfe (FW) algorithm (Pokutta et al., 2020) to train neural networks with constrained parameters on two classification datasets: CIFAR-10 and CIFAR-100

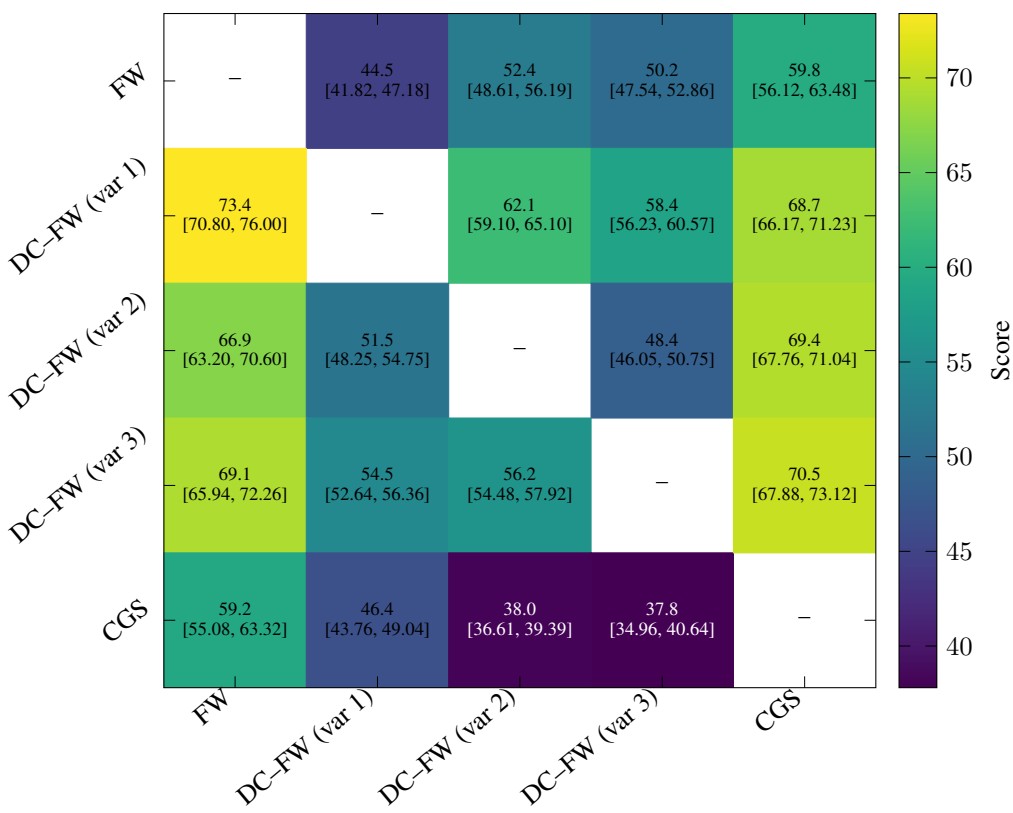

Figure 4: Comparison between FW, DC–FW variants, and CGS with 90% confidence intervals.

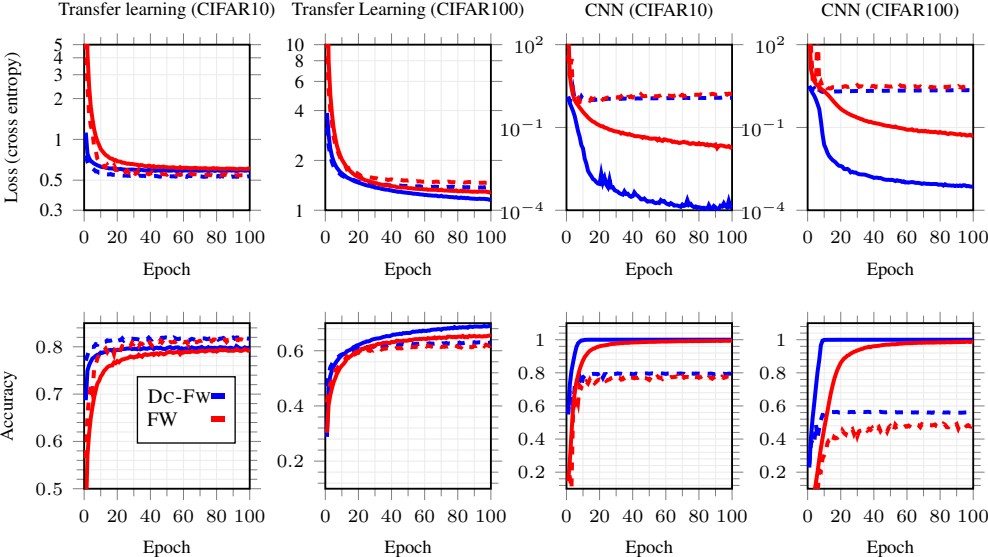

Figure 5: Comparing FW and DC-Fw to train classification task using CE loss using transfer learning on EfficientNetB0 and a customized CNN. The training datasets were CIFAR-10 and CIFAR-100. In all the Figures, dashed and solid lines refer to the validation and the training data, respectively.

(Krizhevsky & Hinton, 2009). The stochastic version of DC-Fw involves random selection of the input data in every iteration of the outer loop. This essentially means that an unbiased estimator of the gradients are used at every iteration.

DC decomposition (9) was used in Dc-Fw with $\phi(x)$ being the empirical loss function using Cross Entropy (CE) for the classification tasks. Note that this decomposition relates to the CGS version of the Dc-Fw algorithm. All experiments were conducted using Python (3.9.5) and PyTorch (2.0.1) on an NVIDIA A100 GPU.

We first trained two customized convolutional neural networks (CNN) for CIFAR-10 and CIFAR-100. Subsequently, EfficientNet-B0 (Tan & Le, 2019) with frozen feature layers was applied to both datasets. All models were initialized using the same seed and trained for 100 epochs with a batch size of 256.

**Structure of the customized convolutional neural networks.** The customized convolutional neural network consists of two convolutional layer blocks followed by a single dense layer block for CIFAR-10, and three convolutional layer blocks followed by a single dense layer block for CIFAR-100. Each convolutional layer block consists of two consecutive convolutional layers with a $3 \times 3$ kernel size, followed by ReLU activations and Batch Normalization. This is followed by a $2 \times 2$ max pooling layer and a dropout layer with a 0.01 probability. This dense layer block starts with flattening the input, followed by two dense linear layers. Each of them uses ReLU activation, Batch Normalization, and Dropout (0.01 probability). The final layer of this block is the classification output layer (dense layer), mapping the extracted high-level features to the desired number of output classes. For CIFAR-10, the network outputs 10 classes, whereas for CIFAR-100, it outputs 100 classes.

**Transfer learning with EfficientNet-B0.** EfficientNet-B0 is the smallest model in the EfficientNet family and serves as a baseline for scaling up to larger variants like EfficientNet-B1 to B7 (Tan & Le, 2019). The detailed structure of EfficientNet-B0 can be found in (Tan & Le, 2019). We applied transfer learning to EfficientNet-B0 by freezing all its feature layers, meaning that the pretrained weights from the model trained on the ImageNet dataset are kept fixed and are not updated during training. Only the classifier layers are trainable, consisting of a dropout layer (with a rate of 0.2) for regularization and a linear layer for classification. This configuration reduces the number of parameters to be trained, thereby saving computational resources while maintaining competitive performance. However, due to the difference between the original domain (features in the ImageNet dataset) and the transferred domain (features in the CIFAR-10/CIFAR-100 dataset), the full potential of EfficientNet-B0 was not fully exploited without trainable feature layers.

**Training setup.** Training was performed with parameters constrained by an $\ell_\infty$-norm ball of radius $c$, following (Pokutta et al., 2020). For the proposed Dc-Fw algorithm, in addition to the breaking condition based on $\text{gap}_{\text{DC}}$ (see Appendix D for more details), an upper bound was applied to the inner loop. This bound was set to 10000 for CNN and 2000 for transfer learning. We further considered the line search for updating $\eta_{t,k}$ to improve convergence.

The CE loss and the top-1 accuracy were the main metrics used for comparison. For the proposed Dc-Fw algorithm and FW algorithm, various choices of $L$ and $c$ were tested. Figure 5 illustrates a performance comparison among different algorithms. The example is based on the hyperparameters $L = 10, c = 10$ for Dc-Fw and FW. Furthermore, Dc-Fw with hyper-parameter $L = 10$, demonstrates superior performance compared to the FW method and in both training and validation.

**Effects of $c$ in neural networks.** The constant scalar $c$ is used to bound the parameters of the network while training. This was shown to be effective in generalization by Pokutta et al. (2020). The value of $c$, however, is dependent on the model and the target dataset and in turn, affects the generalization capabilities of the trained model.

In our numerical simulations, we observed that if $c$ increases, the FW method converges slower. However, Dc-Fw was able to maintain its original performance almost regardless of the value of $c$. Figure 6 depicts an instance of the accuracy of each method in our results. Similar setting as in Section 6 was used and all simulations are for $L = 10$. As shown in the Figure, when $c$ increases, for both datasets and both models, FW becomes slower while Dc-Fw maintains it behavior.

## D   Adaptive Inner-Loop Tolerance Strategy for Dc-Fw

According to Theorem 2, we require $\mathcal{O}(1/\epsilon)$ calls to the subproblem solver (in this case, the FW algorithm) reach an $\epsilon/2$ accuracy. In many practical tasks, this equals a high computational load.

To improve the efficiency of Dc-Fw we used an adaptive tolerance strategy. Suppose we require an accuracy of $\epsilon_0$ for the final solution's gap and each subproblem is solved for an $\epsilon_t$ accuracy. Then, from the proof of Theorem 2, we have

$$f(x_t) - f(x) - \langle \nabla g(x_t), x_t - x \rangle \leq \left( f(x_t) - g(x_t) \right) - \left( f(x_{t+1}) - g(x_{t+1}) \right) + \frac{\epsilon_t}{2}.$$

If we only update $\epsilon_t$ when the problem's gap value at its current iterate is below $\epsilon_t$, then we can write

$$f(x_t) - f(x) - \langle \nabla g(x_t), x_t - x \rangle$$
$$\leq \left( f(x_t) - g(x_t) \right) - \left( f(x_{t+1}) - g(x_{t+1}) \right) + \frac{\max_{x \in \mathcal{D}} f(x_t) - f(x) - \langle \nabla g(x_t), x_t - x \rangle}{2}.$$

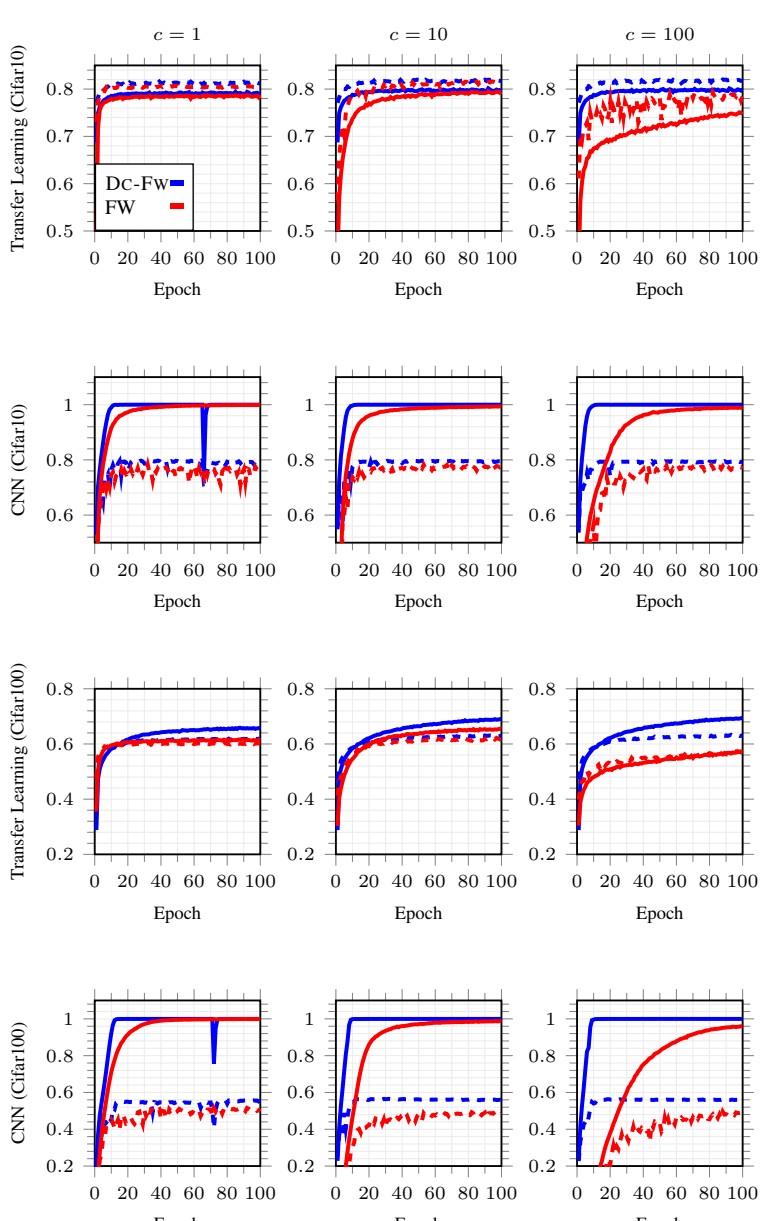

Figure 6: Comparing FW and Dc-Fw to train classification task using CE loss using transfer learning on EfficientNetB0 and a customized CNN for constraint bounds $c = 1, 10, 100$. The training datasets were CIFAR-10 and CIFAR-100. In all the Figures, dashed and solid lines refer to the test and training data, respectively.

Maximizing the left hand side for $x \in \mathcal{D}$ gives

$$\max_{x \in \mathcal{D}} f(x_t) - f(x) - \langle \nabla g(x_t), x_t - x \rangle \leq 2 \left[ \left( f(x_t) - g(x_t) \right) - \left( f(x_{t+1}) - g(x_{t+1}) \right) \right].$$

Now, averaging over $t$ gives the desired inequality below:

$$\min_{0 \leq \tau \leq t} \max_{x \in \mathcal{D}} f(x_\tau) - f(x) - \langle \nabla g(x_\tau), x_\tau - x \rangle \leq 2 \left[ \frac{\left( \phi(x_0) - \phi(x_*) \right)}{t} \right]. \tag{18}$$

Consequently, we update $\epsilon_t$ whenever the gap value falls below it. Here, we assumed a constant multiplier $0 < \beta < 1$ to update $\epsilon_{t+1} = \beta \epsilon_t$. Note however, if this gap falls below $\epsilon_0$, then we terminate the algorithm as the desired accuracy is reached.

With this strategy, $\epsilon_t$ which is also used to terminate the inner loop, can initially take larger values and gradually becomes smaller. In this way, the number of calls to the LMO after $T$ iterations is

$$\sum_{t=1}^{T} \frac{1}{\epsilon_t} \leq T \left( \frac{1}{\min_{0 \leq \tau \leq t} \epsilon_\tau} \right).$$

Also, due to (18), $T$ will be of $\mathcal{O}(1/\epsilon_0)$. Eventually, this implies $\mathcal{O}(\frac{1}{\epsilon_0 \min_{0 \leq \tau \leq t} \epsilon_\tau})$ calls to the LMO.

