# OpenReview forum: "Revisiting Frank-Wolfe for Structured Nonconvex Optimization"
_NeurIPS.cc/2025/Conference — NeurIPS 2025 poster_

### Official Review · Reviewer_ydyb · 2025-06-23

**Clarity:** 2
**Significance:** 3
**Originality:** 2
**Rating:** 4
**Confidence:** 3

**Summary:**

This paper introduces DC-FW, a novel projection-free optimization framework for nonconvex problems formulated as the difference of convex (DC) functions. It combines the classical DCA (Difference-of-Convex Algorithm) structure with the Frank-Wolfe method to solve DC problems without projections. The paper provides theoretical convergence guarantees using a new gap measure (gapDC), and demonstrates that in certain cases, the method achieves improved complexity bounds compared to standard FW. Numerical experiments on QAP, embedding alignment, and neural network training show the versatility and effectiveness of DC-FW.

**Questions:**

1.	While Figure 2 presents FW gap plots against iterations, SVD calls, and wall-clock time, it does not reflect the proposed gapDC measure introduced in the paper. Moreover, the plots do not include theoretical reference lines or convergence slopes (e.g., $O(1/t)$), which would help verify the claimed complexity bounds.
2.	Why are there no comparisons with standard optimizers (e.g., Adam) in the neural network experiments? How does DC-FW perform in such settings?
3.	Can the authors elaborate or visualize the effect of different DC decompositions on performance and convergence?
4.	Please clarify the computational cost per iteration of DC-FW compared to CGS and FW-M, especially in large-scale settings.
5.	Could the authors improve the readability of the QAP results table (e.g., using heatmaps or grouped bar plots with clearer labels)? Additionally, it would be helpful to analyze why DC-FW performs worse than FW on certain QAP instances, to better understand the limitations of the proposed method.
6.	In Figure 4 of the supplementary material, some of the test accuracy curves — particularly for CIFAR-10 under certain constraint values — exhibit sharp downward spikes followed by immediate recovery. Could the authors clarify the cause of this behavior? Is it due to instability in the line search, stochastic subgradient estimation, or numerical issues arising from the projection-free update?

**Ethical Concerns:**

["NO or VERY MINOR ethics concerns only"]

**Final Justification:**

The authors cleared some of my doubts during the rebuttal. Other reviewers also raised good points, and I took due notes.

**Paper Formatting Concerns:**

1.	The discussion of limitations in the conclusion could be moved to a separate “Limitations” section, as recommended by the NeurIPS formatting guidelines.
2.	The theoretical proofs and supplementary experimental results (especially the proofs) currently appear only in the supplementary material. It would be more appropriate to include all proofs as an appendix at the end of the main PDF to ensure better visibility and accessibility for reviewers.

**Quality:**

2

**Strengths And Weaknesses:**

Strengths:
1.	The paper addresses a relevant and challenging class of nonconvex problems in a principled way.
2.	The theoretical contributions are solid, and the proposed gapDC measure is well-motivated and rigorously analyzed.
3.	The framework is general, supports multiple DC decompositions, and includes complexity analysis under different conditions.
4.	Numerical experiments are diverse, covering discrete optimization, matrix problems, and deep learning.
Weaknesses:
1.	The theoretical innovation primarily lies in combining known components (DCA + FW), and is more of a well-engineered synthesis than a fundamentally new algorithm.
2.	The numerical experiments, while varied, lack strong verification of theoretical claims such as complexity bounds (e.g., no clear log-log error vs iteration plots).
3.	The neural network experiments are weakly motivated and lack comparison with standard optimizers (e.g., Adam or SGD).
4.	Presentation of results (especially the QAP table) is dense and difficult to interpret; no standard deviations or statistical significance is reported.

---

> ### Author Rebuttal · Authors · 2025-07-29
>
> We thank the reviewer for their detailed review and thoughtful questions.
>
> **W1:** Our paper is transparent about the connections to existing methods. Our contribution lies in reinterpreting projection-free optimization as a modular framework shaped by the DC structure. This viewpoint highlights that each DC decomposition induces a qualitatively different algorithmic behavior, which is not captured by the standard FW methods. We believe this viewpoint offers a novel and valuable lens and would be of interest to the NeurIPS audience. Please refer to our response to W2 for Reviewer iHJV for some concrete examples of the trade-offs exposed through the DC framework.
>
> **W2:** We appreciate this constructive feedback. The goal of our current experiments is to demonstrate the impact and practical differences between DC-FW variants arising from different DC decompositions. Verifying the theoretical complexity bounds was not the primary focus of our initial experiments. That said, we will include additional experiments to empirically verify the theoretical complexity bounds for completeness (potentially in the supplementary material due to space constraints). We will also include illustrations comparing different notions of gap functions to further clarify the distinctions between convergence criteria.
>
> **W3:** We respectfully disagree. FW has historically not been very successful for neural network training. Our goal in this experiment is not to propose FW as a replacement for optimizers like Adam or SGD. We simply illustrate that by exploiting the right geometric structure, FW can be made to perform reasonably well in a problem where it has been traditionally ineffective. This aligns with the core contribution and main message of our paper. The problem is therefore well-motivated, but our motivation is not to tune FW for large-scale neural network architectures or to benchmark against state-of-the-art optimizers, which would require substantial additional effort beyond the scope of our study.
>
> **W4:** We appreciate this feedback and will improve the presentation of the QAP results. We can also add a table in the appendix to provide complete numerical data. If the reviewer has specific suggestions, we would be happy to incorporate them.
>
> **Q1:** We chose to present the results using the FW gap since it is naturally evaluated by all three methods. For DC-FW, this corresponds to the gap computed in the first inner iteration of each round. While DC-FW also computes the DC gap (approximately, at the final inner iteration), evaluating it for FW-K and FW-M would incur additional computational overhead. Since we report wall-clock time, we did not want to disadvantage the baseline methods by adding this cost. That said, we agree with the reviewer that showing results also with the DC gap would be valuable to verify the theoretical rates. We will conduct these additional experiments for completeness. (We also note that the FW gap is an upper bound on the DC gap, since $f$ is convex.)
>
> It is also worth noting that an $O(1/K)$ rate appears as a straight line with slope -1 in a log-log plot. As seen in Figure 1 [left panel], the rates (in terms of FW gap) for DC-FW and FW-M follow with $O(1/K)$ (where $K$ is the total number of inner iterations for DC-FW), while FW-K exhibits a slightly slower rate. All methods outperform the worst-case $O(1/\sqrt{K})$ bound in this experiment. We will incorporate additional figures (potentially in the supplementary material, due to space constraints) to showcase the empirical convergence rates in detail.
>
> **Q2:** This question relates to W3, and the discussion above addresses it in part. That said, in response to the reviewer’s suggestion, we conducted additional tests. In general, DC-FW performs very similarly to SGD, which is expected given the close connections between the two algorithms under specific DC decompositions.
>
> **Q3:** We appreciate this suggestion, as it truly helps clarifying our main messages in the paper. We have numerically visualized the impact of DC decomposition on gap functions for smooth nonconvex problems in 2D. Unfortunately, due to this year’s rebuttal format, we cannot include figures, but we describe a representative example below.
>
> Consider $\phi(x_1,x_2) = \text{sin}(\pi x_1) ~ \text{cos}(\pi x_2)$ over the domain $[-1,1]^2$, with smoothness constant $L=\pi^2$. We evaluate both the PGM gap and the PPM gap at first-order stationary points (a local minimum, a saddle point, and a local maximum), as well as at small perturbations of these points. We use step size and prox parameter $1/L$. See the table below:
>
> |   x₁   |   x₂   |        Type         |  PGM gap  |  PPM gap  |
> |--------|--------|---------------------|-----------|-----------|
> | -0.50  |  0.00  | Local minimum       |  0.0000   |  0.0000   |
> | -0.49  |  0.01  | Local-min perturbed |  0.0010   |  0.0002   |
> |  0.00  |  0.50  | Saddle              |  0.0000   |  0.0000   |
> |  0.01  |  0.51  | Saddle perturbed    |  0.0010   |  0.1109   |
> |  0.50  |  0.00  | Local maximum       |  0.0000   |  0.0000   |
> |  0.51  |  0.01  | Local-max perturbed |  0.0010   |  0.1110   |
>
> The PGM gap behaves similarly across all stationary points and their perturbations. This is expected because it comes from the decomposition $f(x) = \frac{L}{2}||x||^2$ and $g(x) = \frac{L}{2}||x||^2 - \phi(x)$. Since $g$ is linearized, PGM gap completely discards the curvature information in $\phi$. In contrast, the PPM gap can clearly differentiate between the three types of stationary points. It is relatively flat around the local minimum but sharp near the saddle and local maximum. This behavior arises from the underlying decomposition, $f(x) = \frac{L}{2}||x||^2 + \phi(x)$ and $g(x) = \frac{L}{2}||x||^2$. Since $f$ is not linearized, PPM gap can preserve curvature information in $\phi$.
>
> **Q4:** The overall cost of DC-FW is:
> $$C\_{DC-FW}=\mathcal{O} \left( T_{\epsilon} \left( C\_{\partial g} +  K_{\epsilon} \left( C_{\nabla f} + C_{lmo} + d \right) \right) \right)$$
> where $T_{\epsilon}$ is the number of outer iterations, $K_{\epsilon}$ is the number of inner iterations per outer step, $d$ is the problem dimension, $C_{\partial g}$, $C_{\nabla f}$, and $C_{\text{lmo}}$ denote the cost of computing a subgradient of $g$, a gradient of $f$, and the linear minimization oracle, respectively. Importantly, both $T_{\epsilon}$ and $K_{\epsilon}$ scale with $\epsilon^{-1}$, hence we get
> $$C\_{DC-FW}=\mathcal{O} \left( \epsilon^{-1} C\_{\partial g} +  \epsilon^{-2}  \left( C_{\nabla f} + C_{lmo} + d \right) \right).$$
> It is worth noting that $\epsilon^{-2}$ improves to $ \epsilon^{-3/2}$ if both $\mathcal{D}$ and $f$ are strongly convex.
>
> The cost of FW-K is given by
> $C\_{FW-K}=\mathcal{O} \left( T_{\epsilon}^{FK-K} \left( C\_{\partial g} + C_{\nabla f} + C_{lmo} + d \right) \right)$,
> where $T_{\epsilon}^{FK-K}$ scales with $\epsilon^{-2}$. Hence, the overall cost is
> $$C\_{FW-K}=\mathcal{O} \left( \epsilon^{-2} \left( C\_{\partial g} + C_{\nabla f} + C_{lmo} + d \right) \right).$$
> The cost of FW-M is similar, except for an additional overhead from backtracking line search. Roughly, we have:
> $$C\_{FW-M}=\mathcal{O} \left( \epsilon^{-2} \left( C\_{\partial g} + C_{\nabla f} + C_{lmo} + K_{line-search}(C_{f} + d) \right) \right),$$
> where $K_{\text{line-search}} \geq 1$ is the number of line search steps per iteration, and $C_f$ denotes the cost of evaluating the function $f$. During line search, (sub)gradient and linear minimization oracles are not invoked, but function evaluations and basic arithmetic operations are performed. While line search improves the practical performance of the algorithm, it does not enhance the theoretical guarantees of FW-K in general.
>
> For the specific setting of conditional gradient sliding, that we consider in Section 4.1, we have $f(x) = \frac{L}{2} ||x||^2$ and $g(x) = \frac{L}{2} ||x||^2 - \phi(x)$. Hence, gradient of $f$ comes at the cost of $d$, and the gradient of $g$ comes at the cost of $d + C_{\nabla \phi}$, leading to
> $$C\_{CGS}^{ours} =\mathcal{O} \left( \epsilon^{-1} C_{\nabla \phi} +  \epsilon^{-2}  \left( C_{lmo} + d \right) \right).$$
> This is similar to Qu et al.'s CGS variant, except for the constants. Our CGS strictly improves the constants against Qu et al., specifically, $T_{\epsilon}$ is improved by a factor of $24$. Furthermore, Qu et al. require two FW subroutines (inner loops) per outer iteration, which increases the constants associated with $C_{\text{lmo}}$ and introduces additional arithmetic operations, further increasing the constants in front of $d$.
>
> **Q5:** This question is related to W4 and the discussion above addresses it in part. The current table is in fact similar to a heatmap. We can improve its presentation with clearer labels or by using a heatmap instead. Additionally, we will include a table summarizing the key highlights from this experiment, such as how many times each method achieves the best objective value and relative speed comparisons.
>
> In response to the reviewer’s suggestion, we have also conducted additional experiments. We repeated all QAP experiments over 10 Monte Carlo runs and computed 90% confidence intervals. Due to space limitations, a summary of these extended results is presented in the table in response to "Reviewer KRwt, Limitations".
>
> **Q6:** This occurs only rarely. We note that our current analysis does not account for stochastic gradients, though they are used heuristically in this experiment. Additionally, we employed an adaptive stopping criterion in these experiments. Especially when combined with stochastic gradients, it is possible that the projection subproblem is not solved with sufficient accuracy in some iterations, leading to such unexpected behavior. We believe this issue can be resolved through a more careful analysis that explicitly incorporates gradient noise.

---

> > ### Comment · Reviewer_ydyb · 2025-08-01
> >
> > I thank the reviewers for their detailed explanations. I think this clears a lot of my doubts. I will reconsider the rating in the final phase.

---

> > > ### Author Response · Authors · 2025-08-07
> > > **Thanks**
> > >
> > > Thanks for your feedback and questions, and your willingness to reconsider.

---

### Official Review · Reviewer_iHJV · 2025-07-01

**Clarity:** 2
**Significance:** 2
**Originality:** 2
**Rating:** 4
**Confidence:** 2

**Summary:**

This paper introduces a DC Frank-Wolfe algorithm for DC programming. The proposed method achieves a first-order stationary point in
$O(1/\epsilon^2)$ linear minimization oracle calls. When the set is strongly convex, the oracle complexity improves to $O(1/\epsilon^{1.5})$. The authors also use the proposed method to smooth optimization and specific decompositions can achieve the gradient computation oracle complexity the same as gradient descent.

**Questions:**

1. For the general DC functions, what is the relation bewteen gap_{DC} and $||\hat \partial \phi||$ with $\hat\partial \phi\in \partial \phi$? How to translate gap_{DC} to $||\hat \partial \phi||$ ?

2. What is the benifit of DC structure in nonconvex programming? Can we achieve faster convergence rate than the standard GD or PGD by employing the DC structure?

3. What is the advantage of the DC-FW view compared to the standard analysis framework in Section 4.1?

4. What is the advantage of DC-FW compared to ADMM and its linearized variant.

**Ethical Concerns:**

["NO or VERY MINOR ethics concerns only"]

**Final Justification:**

I have read the rebuttal and the other reviewers' comments. Most of my concerns have been addressed.

**Limitations:**

1. For general DC programming, the proposed method seems a straightforward application of the FW algorithm to DC programming.

2. For the special cases of smooth nonconvex optimization, the DC programming framework seems unnecessary.

**Quality:**

2

**Strengths And Weaknesses:**

Strengths:

1. For general DC programming, this paper introduces a DC Frank-Wolfe algorithm, which achieves a first-order stationary point in
$O(1/\epsilon^2)$ linear minimization oracles. When the set is strongly convex, the calls of linear minimization oracles improve to $O(1/\epsilon^{1.5})$.

2. For the special cases of smooth nonconvex optimization, the proposed DC-FW algorithm with a specific decomposition can achieve a $\epsilon$-suboptimal stationary point measured by $|| x_t - proj_D(x_t-\nabla \phi(x_t)) ||^2\leq gap_{DC}\leq \epsilon$ after $O(1/\epsilon)$ gradient evaluations, which is the same with gradient descent. The proposed DC-FW algorithm is projection-free and it needs $O(1/\epsilon^2)$ linear minimization oracle calls. When the set is strongly convex,  the oracle complexity improves to $O(1/\epsilon^{1.5})$.

Weaknesses:

1. For general DC programming, after reading Section 3, it seems that the proposed method is a straightforward application of the FW algorithm to DC programming. The enhanced performance on strongly convex sets directly applies the theoretical results established in (Garber & Hazan, 2015) and this paper contributes nothing new. The novelty and originality seem below the bar of NeurIPS.

2. For the special cases of smooth nonconvex optimization, I wonder whether the DC programming framework is necessary. For problem (7), we can apply the general projected gradient mehtod with the following iterations:

$x_{t+1}=argmin_{x\in D} \frac{L}{2} || x - (x_t-\frac{1}{L}\nabla \phi(x_t)) ||^2=argmin_{x\in D} \frac{L}{2}||x||^2 - <Lx_t-\nabla \phi(x_t), x>$

When projection is forbidden, we can use the FW algorithm to solve the above problem in the second equation, that is, subproblem (9) in the main paper. Using the standard analysis of inexact projected gradient descent, we can achieve the $O(1/\epsilon)$ gradient oracle complexity (the same with PGD). Using the standard analysis of FW algorithm and the theory in (Garber & Hazan, 2015), we can achieve the $O(1/\epsilon^2)$ and $O(1/\epsilon^{1.5})$ linear minimization oracle complexity, respectively. The DC programming framework seems unnecessary. We can get the same algorithm and achieve the same complexity using existing standard techniques.

3. For the special cases of smooth nonconvex optimization, when using the decomposition in 4.2, the gradient oracle complexity becomes $O(1/\epsilon^2)$, which is higher than the $O(1/\epsilon)$ complexity of PGD.

4. The experiment is inefficient. This paper only compares the posed DC-FW with the original FW algorithm. Suggest also comparing DC-FW with ADMM and its linearized variant (important).

In conclusion, this paper is well-written. However, the theoretical contribution and novelty are limited.

---

> ### Author Rebuttal · Authors · 2025-07-29
>
> We thank the reviewer for their detailed review and thoughtful questions.
>
> **W1:** We respectfully disagree. We believe the reviewer has misinterpreted our contribution as isolated derivations of several projection-free algorithms. In contrast, our work proposes a reinterpretation of projection-free optimization as a modular framework shaped by the DC structure. Each decomposition induces its distinct algorithmic effect, and this perspective is not captured by standard analyses of projection-free methods. We believe this viewpoint offers a novel and valuable lens and would be of interest to the NeurIPS audience.
>
> **W2:** The connection to the projected gradient method is clearly stated in our paper. In lines 187–191, we write:
> >When DCA is applied to this formulation with exact solutions to the subproblems, it recovers the projected gradient method. [...] In other words, DC-FW applied to this formulation simplifies to an inexact projected gradient method, where Frank-Wolfe is used to approximately solve the subproblems. This naturally leads to a nonconvex variant of the conditional gradient sliding algorithm.
>
> We believe this criticism arises from the misunderstanding of our contribution as isolated derivations of several projection-free methods. In contrast, our central message is that the DC framework provides a versatile design tool for constructing and analyzing a broad class of projection-free algorithms. It provides a unified (and surprisingly simple) analysis. More importantly, the guarantees are based on the *DC gap*, which is a generic measure of approximate stationarity (resp., criticality, for nonsmooth problems), and directly reflects the characteristics of the chosen decomposition. This provides insight into which decompositions are better suited in which settings. Since DCA linearizes the $g(x)$ component, the curvature of $g$ is discarded while the curvature of $f$ is preserved. As a result, ill-conditioned terms are better placed in $f$ rather than $g$, to achieve a better notion of stationarity. However, there is a trade-off here, because DC-FW queries the first-order oracle of $f$ more frequently than $g$.
>
> Another trade-off could be the choice of strong convexity in the decomposition. Given any DC decomposition, one can generate a new decomposition with arbitrarily large strong convexity constants simply by adding and subtracting a quadratic term: $f_{\mu}(x) = f(x) + \frac{\mu}{2}||x||^2$, $g_{\mu}(x) = g(x) + \frac{\mu}{2}||x||^2$. The notion of DC gap is strongest when $g$ is merely convex. To see this:
>
> \begin{aligned}
> \text{gap}\_{DC}^{\mu}(x_t)
> &= \max\_{x \in \mathcal{D}} f(x_t) + \frac{\mu}{2}||x_t||^2 - f(x) - \frac{\mu}{2}||x||^2 - \langle \nabla g(x_t) + \mu x_t, x_t - x \rangle \\\\
> &= \max\_{x \in \mathcal{D}}  f(x_t) - f(x) - \langle \nabla g(x_t), x_t - x \rangle - \frac{\mu}{2}||x_t - x||^2  \\\\
> &\leq \max\_{x \in \mathcal{D}} f(x_t) - f(x) - \langle \nabla g(x_t), x_t - x \rangle \\\\
> &= \text{gap}\_{DC} (x_t).
> \end{aligned}
>
> This shows that as we artificially increase strong convexity, DC gap becomes less sensitive to the actual structure of the problem. However, when $\mathcal{D}$ is also strongly convex, we know that strong convexity of $f$ improves the convergence rate for the subproblems. Therefore once again we face a trade-off between computational efficiency and the notion of stationarity.
>
> There is no single universally best method in nonconvex optimization. Our goal is not to propose a one-size-fits-all projection-free method; that would be an unrealistic goal. We instead emphasize the flexibility of the DC decompositions, and how it can be used to balance computational cost and meaningful notions of optimality. Practitioners can try standard FW on nonconvex problems and conclude that it underperforms in their settings. Our work shows that there is more to FW in the nonconvex setting, and that the DC perspective offers a simple and flexible framework for making it work.
>
> **W3:** PGM has convergence guarantees in terms of $\frac{L}{2}||x_k - proj_{\mathcal{D}}(x_k - \frac{1}{L}\nabla \phi(x_k))||^2$. It does not guarantee convergence to a point satisfying $\frac{L}{2}||x_k - prox_{\frac{1}{L}\phi + \iota_\mathcal{D}}(x_k)||^2 \leq \epsilon$ in finite time. The gap between these two quantities can be significant particularly when $\phi$ is ill-conditioned.
>
> **W4:** The comparison with ADMM and its linearized variant is not directly relevant to our setting. Our goal in the experiments is not to compete with proximal or dual-based methods, but to explore how different DC decompositions lead to qualitatively different projection-free algorithms. Therefore, we focus on standard FW methods and DC-FW variants using different decompositions, which is aligned with the core objective of our study. That said, if the reviewer has a specific ADMM variant in mind for a particular experiment with references, we would be happy to include it (potentially in the supplementary material due to space constraints).
>
> **Q1:** Please note that $\partial \phi$ is not always well-defined when $\phi$ is nonconvex, since the subdifferential is only properly defined for convex functions (or defined under more refined notions such as Clarke or limiting subdifferentials). It is important to note that even when $\phi$ is convex but nonsmooth, the subgradient norm $||\hat{\partial} \phi||$ is not a suitable convergence measure. The gradient-norm-based stationarity condition used in smooth nonconvex optimization does not extend meaningfully to nonsmooth settings.  For example, consider the function $\phi(x) = |x|$. Its subdifferential is given by $\text{sign}(x)$ for $x \neq 0$ and the entire interval $[-1, 1]$ at $x = 0$. Although $x = 0$ is the unique stationary point, the subgradient norm remains constant at $1$ as long as $x \neq 0$, regardless of how close we are to the solution. Therefore, subgradient norm cannot be used as a meaningful measure of approximate stationarity in nonsmooth settings.
>
> **Q2/Q3:** These questions are related to W2 and the discussion above addresses them. One additional point worth mentioning is that the DC gap defines a stronger notion of stationarity than the classical projected gradient mapping in Section 4.1. In Lemma 10, we show that
> $$\text{gap}\_{DC}(x_t) \geq \text{gap}\_{PGM}^L(x_t).$$
> This inequality is in fact strict except when both sides are zero (i.e., except when $x_t$ is a stationary point). It is easy to see in the derivation in Appendix A.6, equality holds when
> $$
> \langle \nabla \phi(x_t), x_t - x_t^\star \rangle = L ||x_t - x_t^\star||^2,
> \quad \text{where} \quad x_t^\star = \text{proj}\_{\mathcal{D}} (x_t - \frac{1}{L} \nabla \phi(x_t)),
> $$
> which happens only when $x_t$ minimizes the proximal subproblem, implying $x_t$ is stationary. For any other point, the DC gap is strictly larger than the projected gradient norm.
>
> **Q4:** This question is related to W4 and the discussion above addresses it. We explore how different DC decompositions influence projection-free optimization algorithms within a unified framework. To our knowledge, standard ADMM is not designed for DC optimization and it is not projection-free. If the reviewer has specific variants of ADMM in mind that are particularly relevant to our problem template, we would greatly appreciate references. We are happy to include a discussion (or even additional experiments) if there is a meaningful connection.

---

> ### Comment · Reviewer_iHJV · 2025-08-04
>
> Thank you for your response. I have read through the rebuttal and increased my score from 2 to 3.
>
> Since the scores varied widely, I have re-examined the authors' rebuttal. I acknowledge that this paper's contribution lies in proposing a unified analytical framework. One issue in Weakness 1 remains unresolved. Chapter 3 is the most significant part of the work, but its core relies on two key lemmas cited from prior literature (Lemma 3 and Lemma 5), with limited theoretical proof contributions from the authors—including in the supplementary materials. If the authors can persuade me on this point, I would further raise my score from 3 to 4.

---

> > ### Comment · Reviewer_iHJV · 2025-08-06
> >
> > Kind reminder: The authors have one issue unresolved in the above comment. For a theoretically oriented paper in the optimization field, the limited contribution in theoretical proofs is a notable weakness. Particularly in the supplementary materials there are only two pages for the proofs of the core results of Chapters 3 and 4.

---

> > > ### Author Response · Authors · 2025-08-06
> > >
> > > We thank the reviewer for their engagement in the discussion phase.
> > >
> > > Our contribution should not be viewed merely as a theoretical unification. We propose a modular design framework for projection-free methods in nonconvex optimization, guided by the interpretable choice of DC decomposition. While we previously emphasized this point in our response to W2, it is also directly relevant to W1, as it reflects a key source of novelty and originality in the perspective we introduce. We would like to stress that treating the DC decomposition as a design choice to influence the practical behavior of algorithms, their computational costs, or the nature of the resulting approximate stationary points is not a classical viewpoint in the literature but a key conceptual contribution of our work.
> > >
> > > The reviewer highlights Lemmas 3 and 5 as the core results. However, we would instead emphasize that Definition 1, Lemma 1, and Theorem 2 are more central as they demonstrate that the DC gap provides a suitable measure of stationarity. While the reviewer appears to be expecting a technically intricate theoretical analysis, our contribution lies in offering a novel perspective. We believe it is a mischaracterization to view our submission as an arbitrary combination of DCA and FW and their convergence analysis. While both DCA and FW have been mainstream optimization techniques for the past two to three decades, realizing the value of their integration requires a deep understanding of what each method offers, both in theory and practice.
> > >
> > > Our proposed framework has several immediate implications that we would like to recall:
> > >
> > > 1. We kindly invite the reviewer to check our response to Q3 for Reviewer ydyb, where we demonstrate numerically how different choices of DC decomposition result in varying notions of stationarity. We believe this example helps illustrate the importance of the chosen DC decomposition in shaping the notion of approximate stationarity. In the revision, we will include similar examples along with visual characterizations of the stationarity measures to illustrate the distinctions induced by different DC gaps.
> > >
> > > 2. We would like to emphasize that our proposed framework allows us to transfer all known enhancements of FW from the convex to the nonconvex setting. For instance, Corollary 6 shows how the faster convergence rates of FW over strongly convex domains (established for strongly convex functions) can be leveraged in the nonconvex setting through DCFW. In a similar way, a wide range of results from convex FW can be carried over. (Consider for example the convergence guarantees for FW with away steps over polytopes.) We believe this alone highlights the practical significance of the proposed framework.
> > >
> > > 3. We would like to stress that the FW baseline we compare against in Section 5.1 is taken from the application-focused work of Vogelstein et al. (2015). This well-cited paper demonstrates that standard FW performs well compared to several state-of-the-art methods on QAPLib problems. In our experiments, DCFW considerably outperforms this FW baseline. (Please also see our response to Reviewer KRwt under the “Limitations” section for updated results based on 10 Monte Carlo runs with confidence intervals, extending the original results with a single run in the initial submission.) This comparison provides empirical support for the utility of our algorithmic design framework.
> > >
> > > 4. Previously, Khamaru & Wainwright (2018) and Millan et al. (2023) studied the FW method for DC problems. We attain similar convergence rates (albeit in the DC gap, instead of the classical FW gap) but with reduced oracle complexity for the $g$ component and without relying on complex theoretical arguments. We view the simplicity of our analysis as a strength, not a limitation.
> > >
> > > 5. The prior nonconvex conditional gradient sliding algorithm by Qu et al. (2018) has an intricate theoretical analysis. Our proposed framework offers a simpler solution to the same problem with stronger convergence guarantees, achieved directly through the general DCFW framework, without needing an additional complex analysis. Again, we view the simplicity of our analysis as a strength, not a limitation.
> > >
> > > 6. Our proposed design is flexible and offers new methods. We kindly refer the reviewer to our response to W2 for Reviewer KRwt, where we describe a conditional gradient sliding method for DC functions. Note that the method of Qu et al. (2018) does not apply in this setting, as it requires smoothness of the entire objective, whereas our formulation allows g to be nonsmooth. A concrete example of this setting is the numerical experiment presented in Section 5.2.

---

> > > > ### Comment · Reviewer_iHJV · 2025-08-07
> > > >
> > > > Thanks for the response. Although I still believe that a two-page proof (relatively straightforward) is insufficient for a theoretically oriented paper in the optimization field, I am willing to raise my score. Let time judge the contribution of this paper.

---

> > > > > ### Author Response · Authors · 2025-08-09
> > > > >
> > > > > Thank you for your engagement in the discussion phase and for reconsidering your score.

---

### Official Review · Reviewer_AnH8 · 2025-07-01

**Clarity:** 3
**Significance:** 3
**Originality:** 3
**Rating:** 5
**Confidence:** 4

**Summary:**

In this paper, the authors investigate the Frank-Wolfe algorithm for nonconvex constrained optimization problems with the difference-of-convex structure $f(x) – g(x)$. For this type of problems, a new Frank-Wolfe variant called DC-FW is proposed, which matches the best FW step complexity and achieves a better gradient complexity with respect to function $g(x)$. For strongly-convex feasible domain, it achieves better minimization oracle complexity when the objective functions are strongly-convex or Lipschitz smooth.

**Questions:**

1. Can we split any nonconvex objective function into the dffference of two convex functions? For example, we have a two-layer neural network, how should we convert it to a DC problem?
2. After converting a nonconvex problem to a DC problem, will DC-FW still suffer from the saddle point issue?

**Ethical Concerns:**

["NO or VERY MINOR ethics concerns only"]

**Final Justification:**

My concerns are addressed by the rebuttal. I will retain my score.

**Limitations:**

Yes

**Quality:**

3

**Strengths And Weaknesses:**

Strength

The authors design a novel and efficient variant of Frank-Wolfe algorithm to solve difference-of-convex optimization problems. The new method matches the best FW step complexity and achieves a better gradient complexity with respect to function $g(x)$. For strongly-convex or Lipschitz smooth problems, the minimization oracle complexity can be improved from $O(1/\epsilon^2)$ to $O(1/\epsilon^{\frac{3}{2}})$.

Weakness

Although the generality of the method is clarified in the introduction section, it is not trivial to convert a practical nonconvex loss function to a DC problem, especially for some complicated tasks in CV or NLP.

---

> ### Author Rebuttal · Authors · 2025-07-29
>
> We thank the reviewer for their thoughtful questions.
>
> **W/Q1:** We agree that finding a practical DC decomposition for modern neural architectures used in CV or NLP is a nontrivial task. However, there is growing interest in developing DC decompositions for neural network training. Recent works have already demonstrated such decompositions for multilayer perceptrons and convolutional neural networks [R1, R2], and we expect further progress in this direction.
>
> As for whether any nonconvex objective function can be expressed as a DC function: the strict answer is no. However, DC programming captures a very broad class of nonconvex problems. Verbatim quote from [R3], “DC programs cover most of all nonconvex realistic optimization problems” (see p.6 for a technical discussion). For example, the maximum of a family of twice differentiable functions is DC, any function with locally Lipschitz continuous gradients is DC, the sum or product of two (real-valued) DC functions is also DC. It is important to emphasize, however, that these are existence results. Even when a DC decomposition exists, finding one that is useful for optimization can be challenging.
>
> [R1] Askarizadeh et al., Convex-concave programming: An effective alternative for optimizing shallow neural networks, 2024.
>
> [R2] Awasthi et al., DC-programming for neural network optimizations, 2024.
>
> [R3] Le Thi & Pham Dinh, DC programming and DCA: thirty years of developments, 2018.
>
> **Q2:** This is an interesting question. The short answer is: yes, the method can still suffer from saddle point issues. The DC framework is quite general and also covers standard methods such as gradient descent and the proximal point method as special cases. Since gradient descent is a special case of DCA, all its limitations naturally extend to DCA in general. However, specific DC decompositions may be more or less prone to such issues. The intuition is that DCA retains the function $f$ and linearizes $g$ at each iteration, discarding higher-order information in $g$, both in the algorithm and in the resulting notion of “DC gap” (note that the DC gap is decomposition-dependent). Consequently, a decomposition that retains more of the problem’s geometric structure in $f$ can yield a gap function that more accurately reflects the landscape of the original objective. To illustrate this phenomenon, we conducted a numerical experiment in 2D showing how different DC decompositions affect the behavior of the gap function in smooth nonconvex settings (specifically, we compare the settings from Sections 4.1 and 4.2). We kindly refer the reviewer to our response Q3 for Reviewer ydyb for further details on this example.

---

> > ### Comment · Reviewer_AnH8 · 2025-08-07
> > **Official Comment by AnH8**
> >
> > Thanks for addressing my concerns. I will keep my score.

---

> > > ### Author Response · Authors · 2025-08-07
> > > **Thanks**
> > >
> > > We thank the reviewer for the engagement, feedback, and encouragement.

---

### Official Review · Reviewer_KRwt · 2025-07-02

**Clarity:** 2
**Significance:** 3
**Originality:** 3
**Rating:** 5
**Confidence:** 4

**Summary:**

The paper proposes DC-FW, a projection-free optimization framework for nonconvex problems expressible as the difference of convex (DC) functions. By combining the DC Algorithm (DCA) with Frank-Wolfe (FW) iterations to solve the linearized subproblems, the authors derive an algorithm that maintains the scalability of FW while gaining flexibility from DC decomposition.

**Questions:**

Q1. Could the authors clarify how to choose an optimal DC decomposition for a given objective?

Q2. Is gradient reuse always beneficial?

Q3. Can your method be extended to stochastic settings?

Q4. How does the algorithm behave when $g$ is not differentiable and subgradients are hard to compute?

**Ethical Concerns:**

["NO or VERY MINOR ethics concerns only"]

**Final Justification:**

I thank the authors for nicely and carefully addressing my comments. Considering the contribution and results of this work I raised my score.

**Limitations:**

- Line-search or step-size tuning in practice might be sensitive, especially in large-scale or noisy settings.

- No statistical significance reported for experimental results.

**Paper Formatting Concerns:**

No concerns of paper formatting.

**Quality:**

3

**Strengths And Weaknesses:**

**Strengths**

- A unified view of DC and FW with tight non-asymptotic convergence analysis.
- Accommodates multiple DC decompositions for the same objective.
- Avoids projections—a major bottleneck in constrained optimization.
- Via decomposition that reuses gradients.
- Strong results in classical nonconvex domains.

**Weaknesses**

- Missing real-world deep learning examples: Experiments focus on classical problems.
- Quantitative gains from reusing gradients are not isolated in experiments.

---

> ### Author Rebuttal · Authors · 2025-07-29
>
> We thank the reviewer for their thoughtful comments and questions.
>
> **W1:** In the supplementary material, we presented a simple neural network experiment to illustrate that projection-free methods can perform reasonably well with a suitable DC decomposition. This is interesting because the FW methods have historically not been very successful for neural network training. However, our goal in this experiment is not to position FW as a replacement for state-of-the-art optimizers in large-scale modern applications. Finding the right decompositions for such modern neural architectures in a nontrivial task, and benchmarking against advanced optimizers like Adam, AdamW, or Muon would require substantial additional effort beyond the scope of our study.
>
> **W2:** We appreciate this feedback. As noted in Section 5.2, DC-FW reuses the subgradients of g, resulting in only 88 oracle calls over 10,000 total inner iterations in the embedding alignment experiment. However, we realize we did not report similar results in the conditional gradient sliding setting.
>
> After the submission, we observed that our framework naturally enables gradient sliding for DC functions where $f$ is smooth and $g$ is nonsmooth. This is more general than the smooth setting we considered in Section 4.1. Specifically, since $\frac{L}{2}||x||^2 - f(x)$ is convex for $L$-smooth $f$, we can define a new DC decomposition \phi(x) = f’(x) - g’(x) with
> $$
> f’(x) = \frac{L}{2}||x||^2 \quad \text{and} \quad g’(x) = g(x) + \frac{L}{2}||x||^2 - f(x).
> $$
> Applying DC-FW to this decomposition economizes on both gradient evaluations of $f$ and subgradient evaluations of $g$. This problem setting also covers our experiment in Section 5.2. We will incorporate this variant into this experiment.
>
> **Q1:**  This is an important question, but unfortunately there is no universally optimal way to choose a DC decomposition. Our goal is not to propose a one-size-fits-all projection-free method; such a goal would be unrealistic given the complexity of nonconvex optimization. Our main contribution lies in highlighting how the DC decomposition provides a design space for projection-free methods, where one can balance computational cost and optimality criteria depending on the problem. That said, in certain settings, the trade-offs can be clearly understood; we kindly refer the reviewer to our answer to W2 for Reviewer iHJV, where we discuss two such example cases in detail.
>
> **Q2:** This question is related to the previous one and to our answer to W2 for Reviewer iHJV. The short answer is: *not necessarily*. While the method improves the worst-case gradient oracle complexity for the given problem class, success in nonconvex optimization is not only measured by the rate of convergence, but it also depends on the type of stationary points the method finds. We expect the method to perform well in practice for problems where f is moderately well-conditioned, or in applications such as neural network training where (stochastic) gradient-type methods are empirically known to be effective.
>
> **Q3:** There are at least two natural ways to extend our results to a stochastic setting. First, one can retain the deterministic DCA framework but solve the subproblems using a stochastic FW variant. Second, one can consider a stochastic variant of DCA itself. We note, however, that stochastic DCA variants are less developed in the literature, and existing approaches we are aware of rely on variance-reduction techniques such as SVRG or SAGA. We refer the reviewer to [R1] and [R2] (and references therein) for more details on stochastic FW and stochastic DCA methods, respectively.
>
> [R1] Yurtsever et al., Conditional gradient methods via stochastic path-integrated differential estimator, 2019.
>
> [R2] Le Thi et al., Stochastic DCA with variance reduction and applications in machine learning, 2022.
>
> **Q4:** Since DCA relies on linearizing $g(x)$, it is effective only when its (sub)gradients are accessible. For many common nonsmooth terms such as ReLU, norms, or support function, subgradients are either in closed form or can be efficiently approximated using numerical linear algebra techniques. This is for example the case in Section 5.2, where $g$ is the nuclear norm, and its subgradient can be computed via a singular value decomposition or can be approximated using Newton–Schulz iterations. It is also worth noting that approximate subgradients are available for any prox-friendly function.
>
>
> **Limitations:** In response to the reviewer’s comment on statistical significance, we have also conducted additional experiments. Note that the alignment experiment is a deterministic experiment. We repeated all QAP experiments over 10 Monte Carlo runs and computed 90% confidence intervals (shown in brackets in the form of "**average** [lowerbound upperbound]"). A summary of these extended results is presented in the table below:
>
> | Method     | FW          | DCFW         | DCFW2        | DCFW3       | CGS    |
> |----|-------|---------------|---------------|-----------|----------|
> | **FW**       | 0.0                     | **44.50** [41.82, 47.18]     | **52.40** [48.61, 56.19]     | **50.20** [47.54, 52.86]     | **59.80** [56.12, 63.48]     |
> | **DCFW**     | **73.40** [70.80, 76.00]    | 0.0     | **62.10** [59.10, 65.10]     | **58.40** [56.23, 60.57]     | **68.70** [66.17, 71.23]     |
> | **DCFW2**    | **66.90** [63.20, 70.60]    | **51.50** [48.25, 54.75]     | 0.0         | **48.40** [46.05, 50.75]     | **69.40** [67.76, 71.04]     |
> | **DCFW3**    | **69.10** [65.94, 72.26]    | **54.50** [52.64, 56.36]     | **56.20** [54.48, 57.92]     | 0.0           | **70.50** [67.88, 73.12]     |
> | **CGS**  | **59.20** [55.08, 63.32]    | **46.40** [43.76, 49.04]     | **38.00** [36.61, 39.39]     | **37.80** [34.96, 40.64]     | 0.0       |
>
> The table shows the one on one comparison between the performance of different methods against each other. The inputs represent the average number of datasets (among 10 experiments), that a method on each row wins (achieves lower assignment error) against other methods in each column. For example, DCFW performs better than the CGS method  in **68.70** cases on average with 90% confidence interval [66.17, 71.23].  On the other hand, CGS did better than DCFW in **46.40** cases on average with 90% confidence interval [43.76, 49.04].

---

> > ### Comment · Reviewer_KRwt · 2025-08-04
> > **Answer to authors**
> >
> > Thank you for nicely and carefully addressing my comments. Considering the contribution and results of this work I raised my score.

---

> > > ### Author Response · Authors · 2025-08-05
> > > **Thanks**
> > >
> > > We thank the reviewer for their comments, the discussion, and their appreciation of the work.

---

### Official Review · Reviewer_YETC · 2025-07-03

**Clarity:** 4
**Significance:** 3
**Originality:** 3
**Rating:** 5
**Confidence:** 4

**Summary:**

The article presents a Frank-Wolfe variant (also called projection-free method) DC-FW to minimize a difference of convex functions.
A proof of rate of convergence matching previously known results is presented.
In case of strongly convex constraint set, an improvement for the rate of convergence according to linear minimization oracles is proved.
Two main experiments are conducted to compare the performances of the DC-FW against algorithms found in the literature.

**Questions:**

Here are some questions and remarks I had while reading the paper:
- Line 22-24 Please provide a roof or reference that "any smooth function can be expressed as a DC decomposition".
- In Definition 1, the authors are actually defining a generalized Bregman divergence. See  paragraph 2.2 in  K. C. Kiwiel. _Proximal minimization methods with generalized Bregman functions_. SIAM journal on control and optimization, 35(4):1142–1168, 1997.
-  Line 120 Please add "under assumption of $g$ differentiability"
- Line 126 Maybe write the formula with a $min$
  $$\hat \Phi_t(x_{t+1}) - \hat \Phi_t(x) \leq \epsilon/2, \; \forall x \in \mathcal{D}$$
- Line 135-136 Isn't the lsc assumption implied by $L$-smoothness?
- Line 145-146 "we can always express as a difference of strongly  convex functions by adding the same quadratic term to both f and g. In this case, subproblem " proof or reference?
- Line 168-169 "but rather focus on the classical FW algorithm" Where in the Khamaru & Wainwright reference do we have a mention to FW?

**Ethical Concerns:**

["NO or VERY MINOR ethics concerns only"]

**Final Justification:**

The authors have answered to all my questions and clarified what I've misunderstood. Thus, I retain my current score for the paper.

**Limitations:**

yes

**Paper Formatting Concerns:**

Nothing

**Quality:**

3

**Strengths And Weaknesses:**

**Strengths:**

The paper is well organized and well written.
The setting and contributions are clearly
presented in the introduction.
The paper provides theoretical contribution on the convergence rate
for DC programming using FW over strongly convex set.
The efficiency of DC-FW is demonstrated
through two experiments in paper:
 - Quadratic Assignment Problem (QAP) with  comparisons conducted over **134 datasets**
 - Alignment of Partially Observed Embeddings for 300-dimensional English and French word embeddings.
 The appendix also contains additional experiments with other variants of the algorithm for QAP and additional numerical experiments on neural network training.

**Weaknesses:**

The influence of the choice of the DC decomposition on the convergence rate is not discussed.
In section 4., we are shown two special cases of DC decomposition leading respectively to a gradient sliding problem and proximal point problem for the subproblem of DC-FW. Thus, a  comparison of  between gradient sliding algorithms and proximal point algorithms might be a starting point for a discussion.

---

> ### Author Rebuttal · Authors · 2025-07-30
>
> We thank the reviewer for their detailed review and useful suggestions.
>
> **W:** We appreciate this suggestion. The influence of the DC decomposition on the behavior of the DC gap was central to our thinking while writing the paper, as reflected in the paper’s abstract and main narrative. That said, we agree that demonstrating this explicitly in the specific settings of Sections 4.1 and 4.2 would greatly improve clarity. In response, we have conducted additional numerical experiments on some functions in 2D, where we visualize how different DC decompositions affect the notion of stationarity captured by the DC gap. Unfortunately, due to this year’s rebuttal format, we are unable to upload figures. However, we have included a table that showcases our main observations from one such example, and we kindly refer the reviewer to our answer to Q3 for Reviewer ydyb for further details on this example.
>
> **Q1:** Let $\phi : \mathbb{R}^n \to \mathbb{R}$ be an $L$-smooth function:
> $$
> || \nabla \phi(x) - \nabla \phi(y) || \leq L ||x - y||, \quad \forall x,y \in \mathbb{R}^n.
> $$
>
> Then, $\hat{\phi}(x) = \frac{L}{2}||x||^2 - \phi(x)$ is convex, since $\forall x,y \in \mathbb{R}^n$, we have
> \begin{aligned}
> \langle \nabla \hat{\phi}(x) - \nabla \hat{\phi}(y), x - y \rangle
> & = L ||x-y||^2 - \langle \nabla \phi(x) - \nabla \phi(y), x - y \rangle \\\\
> & \geq L ||x-y||^2 - || \nabla \phi(x) - \nabla \phi(y) || \cdot || x - y || \\\\
> & \geq L ||x-y||^2 - L || x - y||^2 \\\\
> & = 0
> \end{aligned}
> where the first inequality is due to Cauchy-Schwarz, and the second one follows from the smoothness of $\phi$.
>
> We can show convexity of $\frac{L}{2}||x||^2 + \phi(x)$ in a similar way.
> However, this follows immediately from the observation that smoothness is preserved under negation: If $\phi$ is $L$-smooth, then $-\phi(x)$ is $L$-smooth as well. Applying the previous result to $-\phi(x)$, we conclude that $\bar{\phi}(x) = \frac{L}{2}||x||^2 - (-\phi(x)) = \frac{L}{2}||x||^2 + \phi(x)$ is also convex.
>
> **Q2:** The generalized Bregman Divergence as in (Kiwiel, 1997) is defined as
>
> $$D_h(x,y) = h(x) - h(y) - \langle u ,x-y \rangle , \quad u \in \partial h.$$
>
> Although it looks similar to the gap function in Definition 1, the main difference lies in the subgradient component. Our subgradient belongs to another function and not $h$. For example, if the DC decomposition is $\phi(x) = h(x) - g(x)$, our gap becomes
> $$gap(x) =\max_y \left\\{ h(x) - h(y) - \langle u ,x-y \rangle \right\\} , \quad u \in \partial g.$$
>
> **Q3:** We will rephrase this sentence as suggested to avoid confusion.
>
> **Q4:** We will rewrite the formula using min as suggested.
>
> **Q5:** Yes it does, thank you for pointing out. We will remove the redundant assumption.
>
> **Q6:** A standard definition of strong convexity states that a function $\psi : \mathbb{R}^n \to \mathbb{R}$ is $\mu$-strongly convex (with respect to the Euclidean norm) if and only if $\psi(x) - \frac{\mu}{2}||x||^2$ is convex. Now suppose $\phi(x) = f(x) - g(x)$ is a DC decomposition, hence $f$ and $g$ are convex. Then, we can define:
> $$f’(x) = f(x) + \frac{\mu}{2}||x||^2 \quad \text{and} \quad g’(x) = g(x) + \frac{\mu}{2}||x||^2.$$
> This gives another DC decomposition $\phi(x) = f’(x) - g’(x)$ where both $f’$ and $g’$ are $\mu$-strongly convex by construction.
>
> **Q7:** Thank you for your careful reading and for raising this point. Your question led us to realize that there are two versions of Khamaru & Wainwright's paper, one presented at ICML and a longer version published in JMLR. The Frank-Wolfe method appears only in the JMLR version (as Algorithm 3, in Section 2.5), whereas we mistakenly cited the ICML version. We will correct this reference.

---

> > ### Comment · Reviewer_YETC · 2025-08-04
> >
> > I thank the authors for their precise and well-argued answers.
> > Indeed, I have missed the $g$ in Definition 1 of the gap, thank you for the clarification.
> > I have appreciated your answers to Q1 and Q6 and hope to see them integrated in the paper in some way.
> > I retain my current score of 5 (Accept).

---

> > > ### Author Response · Authors · 2025-08-05
> > > **Thanks**
> > >
> > > We thank the reviewer for their comments, insights, suggestions, and support.

---

### Note · Authors · 2025-08-15

Dear all,

We feel that our work was received very positively. During the discussions, we reiterated that our contribution should not be viewed only as an algorithm for solving DC problems. The design perspective enabled by the DC decomposition is central to our work, as articulated in our abstract:

> DC decompositions are not unique; by carefully selecting a decomposition, we can better exploit the problem structure, improve computational efficiency, and adapt to the underlying problem geometry to find better local solutions.

In light of this, we summarize our contributions and their implications:

**DC-gap.** We introduce a decomposition-dependent gap function to measure first-order stationarity. The DC-gap linearizes $g$ while preserving $f$, resulting in a flexible and modular stationarity notion shaped by the geometry of $f$ and the first-order structure of $g$, while discarding higher-order information.

**Convergence guarantees.** DCFW reaches an $\epsilon$-solution in $O(1/\epsilon)$ outer iterations, amounting to $O(1/\epsilon^2)$ FW steps (each outer iteration solves a convex subproblem via inner FW steps). This matches the $O(1/\epsilon^2)$ rate of nonconvex FW (albeit in terms of DC-gap). Unlike classical FW, which relies on a fixed linearization, DCFW introduces modularity through decomposition choice.

**DC for algorithmic design.** The decomposition directly shapes the optimization path, enabling trade-offs between stationarity quality and oracle complexity. To illustrate, we introduced two tailored decompositions for smooth minimization: one leads to a conditional gradient sliding (CGS) variant with guarantees in terms of the projected gradient mapping, and the other yields guarantees based on the proximal point mapping. We also extended the CGS variant to nonsmooth problems of the form (smooth f – nonsmooth g) in the discussions.

**Improved rates under additional assumptions.** DCFW inherits known enhancements of FW from the convex to the nonconvex setting. For instance, it achieves an improved rate of $O(1/\epsilon^{3/2})$ FW steps over strongly convex domains, and even faster rates with away steps over polyhedral sets.

**Numerical experiments.** We demonstrate the superior performance of DCFW (with suitable, tailored decompositions) by comparing it against other FW variants.

Together, these contributions demonstrate how DC decompositions can be leveraged as a powerful design tool in projection-free optimization.

---

### Decision · Program_Chairs · 2025-09-17

**Decision:**

Accept (poster)

**Comment:**

**Summary:**
The paper considers constrained optimization problem whose objective is the difference of convex functions, classic Frank-Wolfe (FW) method and difference-of-convex algorithm are adopted which lead to the proposed DC-FW method. Theoretical analysis of the proposed scheme under different condition of the constraint set is provided, and numerical experiments are presented to validate the performance.

**Strength:**
 - A novel and efficient variant of FW method to solve difference-of-convex optimization problem with constraint.
 - Theoretical guarantees with match the existing complexity bound.
 - Solid numerical experiments.

**Weakness:**
 - The presentation of the paper can be improved for better clarity, including 1) discussions on comparison with existing approaches; 2) needs better explained motivation and challenges; 3) numerics can be further strengthened.

**Reason of decision:**
The topic of the paper and its presented result have potential broad interests to the community. During the rebuttal phase, most concerns from the reviewers are fairly addressed. However, incorporating them into the final version of the paper needs substantial work and the authors are recommended to make a thorough revision.